# PIP$_2$ depletion promotes TRPV4 channel activity in mouse brain capillary endothelial cells

Osama F Harraz[1], Thomas A Longden[1], David Hill-Eubanks[1], Mark T Nelson[1,2]*

[1]Department of Pharmacology, University of Vermont, Burlington, United States; [2]Institute of Cardiovascular Sciences, Manchester, United Kingdom

**Abstract** We recently reported that the inward-rectifier Kir2.1 channel in brain capillary endothelial cells (cECs) plays a major role in neurovascular coupling (NVC) by mediating a neuronal activity-dependent, propagating vasodilatory (hyperpolarizing) signal. We further demonstrated that Kir2.1 activity is suppressed by depletion of plasma membrane phosphatidylinositol 4,5-bisphosphate (PIP$_2$). Whether cECs express depolarizing channels that intersect with Kir2.1-mediated signaling remains unknown. Here, we report that Ca$^{2+}$/Na$^+$-permeable TRPV4 (transient receptor potential vanilloid 4) channels are expressed in cECs and are tonically inhibited by PIP$_2$. We further demonstrate that depletion of PIP$_2$ by agonists, including putative NVC mediators, that promote PIP$_2$ hydrolysis by signaling through G$_q$-protein-coupled receptors (G$_q$PCRs) caused simultaneous disinhibition of TRPV4 channels and suppression of Kir2.1 channels. These findings collectively support the concept that G$_q$PCR activation functions as a molecular switch to favor capillary TRPV4 activity over Kir2.1 signaling, an observation with potentially profound significance for the control of cerebral blood flow.

DOI: https://doi.org/10.7554/eLife.38689.001

## Introduction

Endothelial cells (ECs) line the lumen of all blood vessels and are important regulators of artery and arteriole smooth muscle contractility. In capillaries, which lack an overlying smooth muscle cell layer, this regulatory function is absent. In these smallest of all blood vessels, ECs instead support the nutrient- and gas-exchange function characteristic of capillary beds generally. In the brain, capillary ECs (cECs) also constitute the blood-brain barrier, reflecting the presence of tight junctions between adjacent cECs. Not surprisingly given these differing cellular missions, the molecular repertoire of brain cECs and arterial/arteriolar ECs exhibit some marked differences. Notably, intermediate and small-conductance Ca$^{2+}$-activated K$^+$ channels (IK and SK, respectively), which are important in transducing elevations in intracellular Ca$^{2+}$ concentration ([Ca$^{2+}$]$_i$) into membrane potential hyperpolarization in artery/arteriolar ECs to cause relaxation of electrically coupled smooth muscle cells (*Ledoux et al., 2008*; *Sonkusare et al., 2012*; *Taylor et al., 2003*), are absent in brain cECs (*Longden et al., 2017*). However, signaling through G protein–coupled receptors of the G$\alpha_{q/11}$-subtype (G$_q$PCRs)—an important component of this smooth muscle regulatory axis—is robust in brain cECs (*Harraz et al., 2018*). In the canonical pathway of G$_q$PCR signaling, G$\alpha_q$ released upon agonist binding activates phospholipase C (PLC), which hydrolyzes the minor inner leaflet phospholipid, phosphatidylinositol 4,5-bisphosphate (PIP$_2$). Breakdown of PIP$_2$ yields diacylglycerol (DAG) and inositol 1,4,5-trisphosphate (IP$_3$), the latter of which increases [Ca$^{2+}$]$_i$ by acting on its cognate receptor (IP$_3$R) on the endoplasmic reticulum (ER) to promote Ca$^{2+}$ release from intracellular stores.

Blood delivery within the brain is mediated by an ever-narrowing vascular tree comprising surface arteries, penetrating (parenchymal) arterioles and a vast network of capillaries, which enormously

*For correspondence: Mark.Nelson@uvm.edu

Competing interests: The authors declare that no competing interests exist.

**eLife digest** Capillaries form branching networks that surround all cells of the body. They allow oxygen and nutrient exchange between blood and tissue, but this is not their only role. Capillaries in the brain form a tight barrier that prevents components carried in the blood from easily reaching the brain compartment. They also detect the activity of neurons and trigger on-demand increases in blood flow to active regions of the brain. This role, revealed only recently, depends upon ion channels on the surface of the capillary cells. Active neurons release potassium ions, which open a type of ion channel called Kir2.1 that allows potassium inside the cell to flow out. This process is repeated in neighboring capillary cells until it reaches an upstream vessel, where it causes the vessel to relax and increase the blood flow.

Kir2.1 channels sit astride the membranes of capillary cells, where they can interact with other membrane molecules. One such molecule, called $PIP_2$, plays several roles in relaying signals from the outside to the inside of cells. It also physically interacts with channels in the membrane, including Kir2.1 channels. If $PIP_2$ levels are low, Kir2.1 channel activity decreases.

Here, Harraz et al. discovered that capillary cells contain another type of ion channel, called TRPV4, which is also regulated by $PIP_2$. But unlike Kir2.1, its activity increases when $PIP_2$ levels drop. Moreover, TRPV4 channels allow sodium and calcium ions to flow into the cell, which has an effect opposite to that of potassium flowing out of the cell.

Capillary cells also have receptor proteins called $G_qPCRs$ that are activated by chemical signals released by active neurons in the brain. $G_qPCRs$ break down $PIP_2$, so their activity turns Kir2.1 channels off and TRPV4 channels on. This resets the system so that it is ready to respond to new signals from active neurons. $G_qPCRs$ work as molecular switches to control the balance between Kir2.1 and TRPV4 channels and turn brain blood flow up and down.

$G_qPCRs$ and ion channels that depend on $PIP_2$ can also be found in other types of cells. These findings could reveal clues about how signals are switched on and off in different cells. Understanding the role of $PIP_2$ in signaling could also unveil what happens when signaling go wrong.

DOI: https://doi.org/10.7554/eLife.38689.002

extend the territory of perfusion (*Blinder et al., 2013*). Because the brain lacks substantial energy reserves, it relies on an on-demand mechanism for redistributing oxygen and nutrients to regions of higher neuronal activity, a process in which products released by active neurons trigger a local increase in blood flow. This use-dependent increase in local blood flow (functional hyperemia), mediated by a process termed neurovascular coupling (NVC), is essential for normal brain function (*Iadecola and Nedergaard, 2007*) and represents the physiological basis for functional magnetic resonance imaging (*Raichle and Mintun, 2006*).

We recently reported that brain capillaries act as a neuronal activity-sensing network, demonstrating that brain cECs are capable of initiating an electrical (hyperpolarizing) signal in response to neuronal activity that rapidly propagates upstream to dilate feeding parenchymal arterioles and increase blood flow locally at the site of signal initiation (*Longden et al., 2017*). We also established the molecular mechanism underlying this process, showing that extracellular $K^+$—a byproduct of every neuronal action potential—is the critical mediator and identifying the Kir2.1 channel as the key molecular player (*Longden et al., 2017*). Kir2.1 channels are activated by external $K^+$ and exhibit steep activation by hyperpolarization (*Hibino et al., 2010*; *Longden and Nelson, 2015*), characteristics that facilitate regenerative $K^+$ conductance in neighboring cECs and enable long-range propagation of the hyperpolarizing signal (*Longden et al., 2017*). Intriguingly, we have since discovered that decreases in capillary endothelial $PIP_2$ levels induced by $G_qPCRs$ agonists suppress Kir2.1 channel currents, reflecting the fact that $PIP_2$ is required for Kir2.1 channel activity (*Harraz et al., 2018*). Importantly, this suppressive effect of $PIP_2$ depletion exerts a major modulatory influence on $K^+$-evoked hyperemic responses in vivo. In addition to regulating Kir2 family members, $PIP_2$ has been shown to bind to and regulate a diverse array of ion channels (*Hille et al., 2015*). Notable in this context, $PIP_2$ binding has been shown to inhibit several members of the transient receptor potential (TRP) family of non-selective cation channels, including the vanilloid subtype TRPV4 (*Nilius et al.,*

*2008*; *Rohacs, 2009*; *Takahashi et al., 2014*), which we and others have shown is an important $Ca^{2+}$ influx pathway in arterial and arteriolar ECs (*Earley and Brayden, 2015*; *Sonkusare et al., 2014*; *Sonkusare et al., 2012*; *White et al., 2016*).

The Kir2.1 channel appears to be the major $K^+$ channel type in brain cECs (*Longden et al., 2017*). However, the identity of depolarizing ($Na^+/Ca^{2+}$-permeable) channels in cECs is not known. Here, we show that TRPV4 channels are present in cECs and exhibit an exceedingly low open probability under basal conditions due to tonic inhibition by $PIP_2$. We have further found that this inhibition is relieved through $PIP_2$ depletion, independent of the action of the $PIP_2$ hydrolysis products, DAG and $IP_3$. These findings provide a counterpoint to our recent demonstration that Kir2.1 channel activity is sustained by basal levels of $PIP_2$ and suppressed by $G_qPCR$-mediated $PIP_2$ depletion (*Harraz et al., 2018*). Collectively, our findings indicate that $PIP_2$ exerts opposite effects on TRPV4 and Kir2.1 channels in brain cECs, demonstrating that a single regulatory pathway governs the balance between two divergent signaling modalities—one depolarizing and the other hyperpolarizing—in the capillary endothelium.

## Results

### TRPV4 channels in brain cECs are inhibited by intracellular ATP

We previously reported that the selective TRPV4 agonist GSK1016790A (hereafter GSK101) is a potent activator of TRPV4 currents in mesenteric artery ECs that induces maximal arterial dilation at concentrations in the low nanomolar range (*Sonkusare et al., 2014*; *Sonkusare et al., 2012*). To assess TRPV4 channel expression and function in brain cECs, we first performed patch-clamp electrophysiology experiments on freshly isolated cECs from C57BL/6J mouse brain slices, prepared as described in Materials and methods. Specifically, we recorded outward $K^+$ currents mediated by TRPV4 channels in the cytoplasm-intact, perforated whole-cell configuration using a 300 ms ramp protocol (−100 to +100 mV, from a holding potential of −50 mV). Recordings were made in the presence of the voltage-dependent pore blocker ruthenium red (RuR; 1 µM), an approach we have previously used to assess TRPV4 currents in mesenteric artery ECs (*Sonkusare et al., 2012*). Under these conditions, depolarizing ramps rapidly displace RuR, allowing unimpeded outward $K^+$ currents to be detected while preventing $Na^+/Ca^{2+}$ influx and associated $Ca^{2+}$ overload and cell death (*Sonkusare et al., 2012*). Unexpectedly, we found that GSK101 was unable to evoke a detectable whole-cell TRPV4 current in cECs at concentrations that maximally activate TRPV4 channels in mesenteric artery ECs; in fact, even GSK101 concentrations as high as 100 nM failed to stimulate a detectable TRPV4 current in isolated cECs (*Figure 1A*). However, we serendipitously found that breaking through the cell membrane from a perforated patch in the presence of an otherwise ineffective GSK101 concentration (100 nM) resulted in the gradual (~5 min) development of a robust TRPV4 current, suggesting that a factor that suppresses TRPV4 channels had been washed out of the cell.

In the conventional whole-cell configuration with physiological levels of Mg-ATP (1 mM) included in the patch pipette, application of 100 nM GSK101 also failed to evoke an outward current. In striking contrast, this same concentration of GSK101 caused the gradual (~4–5 min) development of a robust outward current in the conventional whole-cell configuration when Mg-ATP was omitted from the pipette solution (*Figure 1A*). Collectively, these observations suggest that ATP is the suppressive factor in question. Consistent with this interpretation, intracellular Mg-ATP (1 mM) caused a dramatic shift in the sensitivity of cEC TRPV4 channels to GSK101: at a potential of 100 mV, the $EC_{50}$ of GSK101 for TRPV4 channels in the presence of Mg-ATP was 415 nM. However, in the absence of Mg-ATP, the $EC_{50}$ was reduced ~23 fold to 18 nM (*Figure 1C*; *Figure 1—figure supplement 1A,B*), a value comparable to that observed in mesenteric artery ECs (*Sonkusare et al., 2014*). The suppressive effect of ATP was specific for adenosine nucleotides, as inclusion of 1 mM GTP in the patch pipette did not alter currents (*Figure 1—figure supplement 1C*). GSK101-induced currents were completely abolished by the selective TRPV4 channel antagonist HC-067047 (1 µM) and were absent in cECs from TRPV4-knockout ($TRPV4^{-/-}$) mice (*Figure 1B*; *Figure 1—source data 1*), confirming that these currents are attributable to TRPV4 channels. The inhibitory effect of intracellular Mg-ATP (1 mM) was not limited to GSK101-activated TRPV4 currents. As shown (*Figure 1—figure supplement*

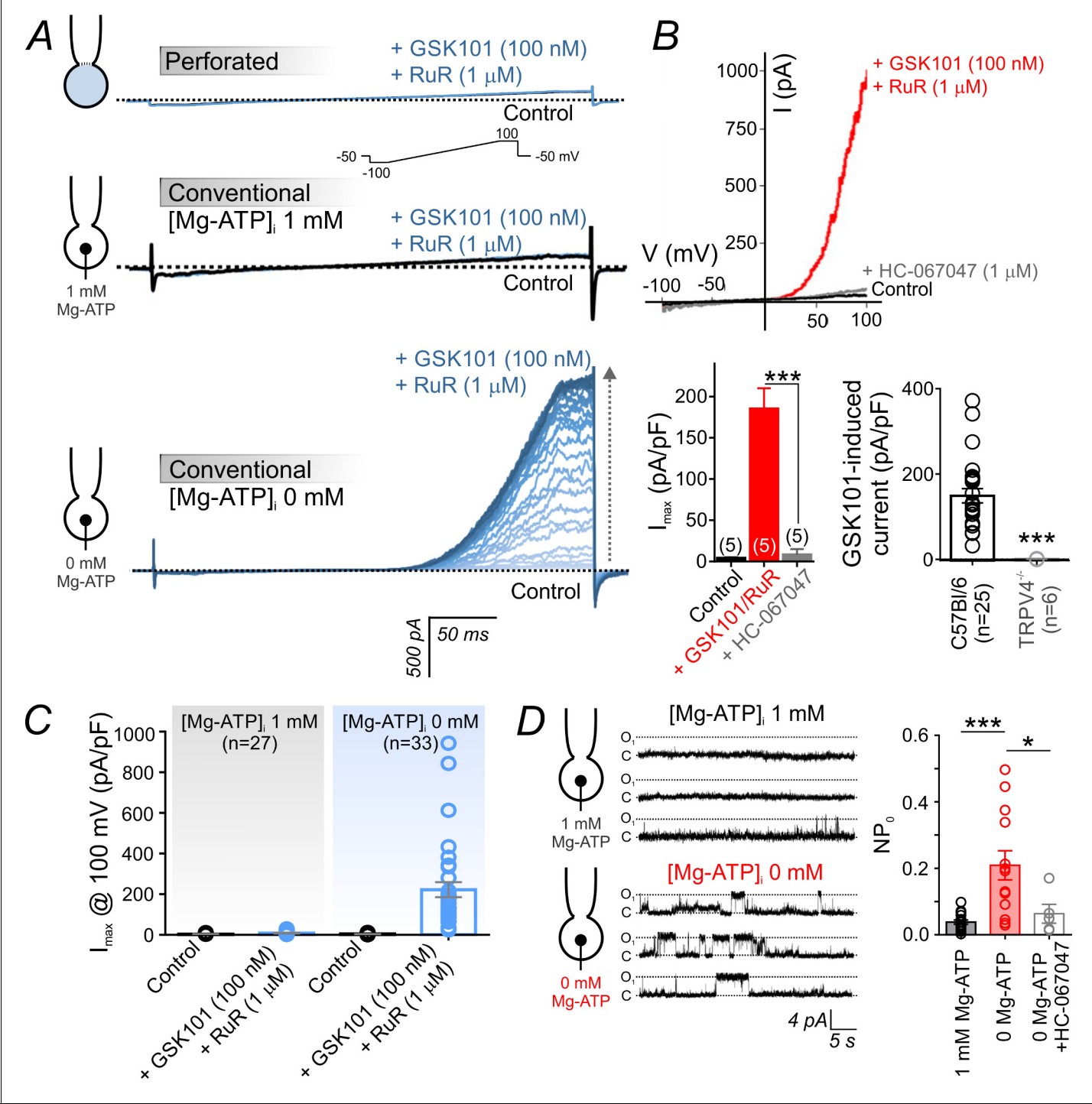

**Figure 1.** Intracellular ATP suppresses TRPV4 currents in cECs. (**A**) Representative traces of TRPV4 currents recorded from freshly isolated cECs using voltage ramps (−100 to 100 mV, from a holding potential −50 mV; inset) and different patch-clamp configurations before (black) and after (blue) the application of 100 nM GSK101 and 1 µM RuR. *Top*: Currents recorded in the perforated whole-cell configuration. *Middle*: Currents recorded in the conventional whole-cell configuration (dialyzed cytoplasm, 1 mM Mg-ATP in the pipette solution). *Bottom*: Currents recorded in the conventional whole-cell configuration (0 mM Mg-ATP in the pipette solution) developed gradually over ~4 min. (**B**) Current-voltage relationship (*top*) and summary data (*bottom left*) of currents recorded before (control) and after the cumulative application of GSK101 (100 nM)+RuR (1 µM) followed by HC-067047 (1 µM) (means ± SEM, ***p<0.001, unpaired Student's t-test; n = 5 each). *Bottom right*: Individual-value plot of peak outward GSK101 (100 nM)-induced currents in cECs isolated from brains of C57Bl/6 (n = 25) or TRPV4$^{-/-}$ (n = 6) mice. A minimum duration of ~5 min after the application of GSK101 was allowed for outward TRPV4 current to develop in each cEC. Data are presented as means ± SEM (***p<0.001, unpaired Student's t-test). (**C**) Individual-

*Figure 1 continued on next page*

*Figure 1 continued*

value plot of peak outward currents recorded at 100 mV before and after the application of GSK101 (100 nM) onto cECs dialyzed with 0 or 1 mM Mg-ATP in the pipette solution. Individual data points are shown together with means (column bars) and SEM (error bars). (D) Representative traces (*left*) and summary individual-value plot (*right*) of TRPV4 single-channel activity. Single-channel openings of TRPV4 channels were recorded as outward quantal $K^+$ currents from cECs in the absence of GSK101 (conventional whole-cell configuration; holding potential, +50 mV; sampling rate, 20 kHz; low-pass filter frequency, 1 kHz; average recording time for each data point, 6 min). cECs were dialyzed with 0 mM (n = 13) or 1 mM (n = 16) Mg-ATP. One group of cECs dialyzed with 0 mM Mg-ATP was treated with 1 μM HC-067047 (n = 5). Data are presented as means (column bars) ± SEM (error bars; *p<0.05, ***p<0.001, one-way ANOVA followed by Tukey's multiple comparisons test).

DOI: https://doi.org/10.7554/eLife.38689.003

The following source data and figure supplements are available for figure 1:

**Source data 1.** Numerical data that were used to generate the chart in *Figure 1B*.
DOI: https://doi.org/10.7554/eLife.38689.007
**Source data 2.** Numerical data that were used to generate the chart in *Figure 1C*.
DOI: https://doi.org/10.7554/eLife.38689.008
**Source data 3.** Numerical data that were used to generate the chart in *Figure 1D*.
DOI: https://doi.org/10.7554/eLife.38689.009
**Figure supplement 1.** Intracellular ATP suppresses TRPV4 activity in cECs.
DOI: https://doi.org/10.7554/eLife.38689.004
**Figure supplement 2.** 4-α-PDD-induced current in cECs.
DOI: https://doi.org/10.7554/eLife.38689.005
**Figure supplement 3.** Single-channel TRPV4 currents in cECs.
DOI: https://doi.org/10.7554/eLife.38689.006

*2*), it also suppressed both inward and outward currents in cECs activated by the TRPV4 agonist 4α-phorbol 12,13-didecanoate (4α-PDD; 5 μM, in the absence of RuR).

There was no detectable whole-cell TRPV4 current in the absence of a TRPV4 activator (*Figure 1A*), suggesting that these channels have a very low basal open probability. To further investigate this, we monitored single TRPV4 channel openings using dialyzed cECs in the conventional whole-cell configuration, an approach that should allow detection of openings of all channels throughout the cEC plasma membrane while controlling intracellular composition. First, we measured TRPV4 unitary currents in dialyzed cECs (0 mM Mg-ATP) held at +50 mV before and after the application of 100 nM GSK101. The unitary current amplitude in the presence of GSK101, estimated from amplitude histograms, was 4.6 ± 0.3 pA (n = 3) (*Figure 1—figure supplement 3A*). This value corresponds to a single-channel conductance of 92 ± 6 pS, in line with values reported in an earlier investigation of TRPV4 channels in native arterial ECs (*Watanabe et al., 2002*) and heterologous expression systems (*Loukin et al., 2010*). Extending these results, we next performed a series of single-channel recordings in the absence of GSK101. cECs held at +50 mV and dialyzed with Mg-ATP (1 mM) displayed infrequent openings ($NP_O$ = 0.038 ± 0.006), but removing ATP increased single-channel opening by ~6 fold ($NP_O$ = 0.209 ± 0.044) (*Figure 1D*; *Figure 1—source data 3*). Single-channel openings in the absence of agonist exhibited a unitary current of 4.5 ± 0.3 pA at +50 mV (*Figure 1D*, *Figure 1—figure supplement 3B*), similar to that evoked using a specific TRPV4 agonist (4.6 pA) (*Figure 1—figure supplement 3A*) and corresponding to a conductance of 90 ± 6 pS. Single-channel currents recorded in the absence of GSK101 were inhibited by the TRPV4 blocker HC-067047 ($NP_O$ = 0.063 ± 0.028) (*Figure 1D*; *Figure 1—source data 3*), confirming their identity as TRPV4 channel-mediated currents. Based on our estimate of ~225 functional TRPV4 channels per cEC (see Materials and methods), we calculate that capillary TRPV4 channels have a basal open probability ($P_O$) of ~0.0002. These observations collectively indicate that capillary TRPV4 channels exhibit a low open probability under basal conditions, and that both constitutive and agonist-induced TRPV4 activities are suppressed by intracellular ATP and recover following ATP washout/dialysis.

## TRPV4 channel suppression is dependent on ATP hydrolysis, but independent of protein kinase A, G or C

The $Mg^{2+}$ salt of ATP is readily hydrolyzable and can be utilized by kinases (*O'Rourke, 1993*). To test whether TRPV4 suppression requires ATP hydrolysis, we replaced ATP in the patch pipette with

the poorly hydrolyzable analog, ATP-γ-S (1 mM). Under these conditions, GSK101 (100 nM) increased TRPV4 currents to same extent as observed in cells dialyzed with 0 mM Mg-ATP, indicating that ATP hydrolysis is required for channel inhibition (*Figure 2A,B*; *Figure 2—source data 1*). This implies an energy-consuming process, and suggests the involvement of an enzyme, possibly a

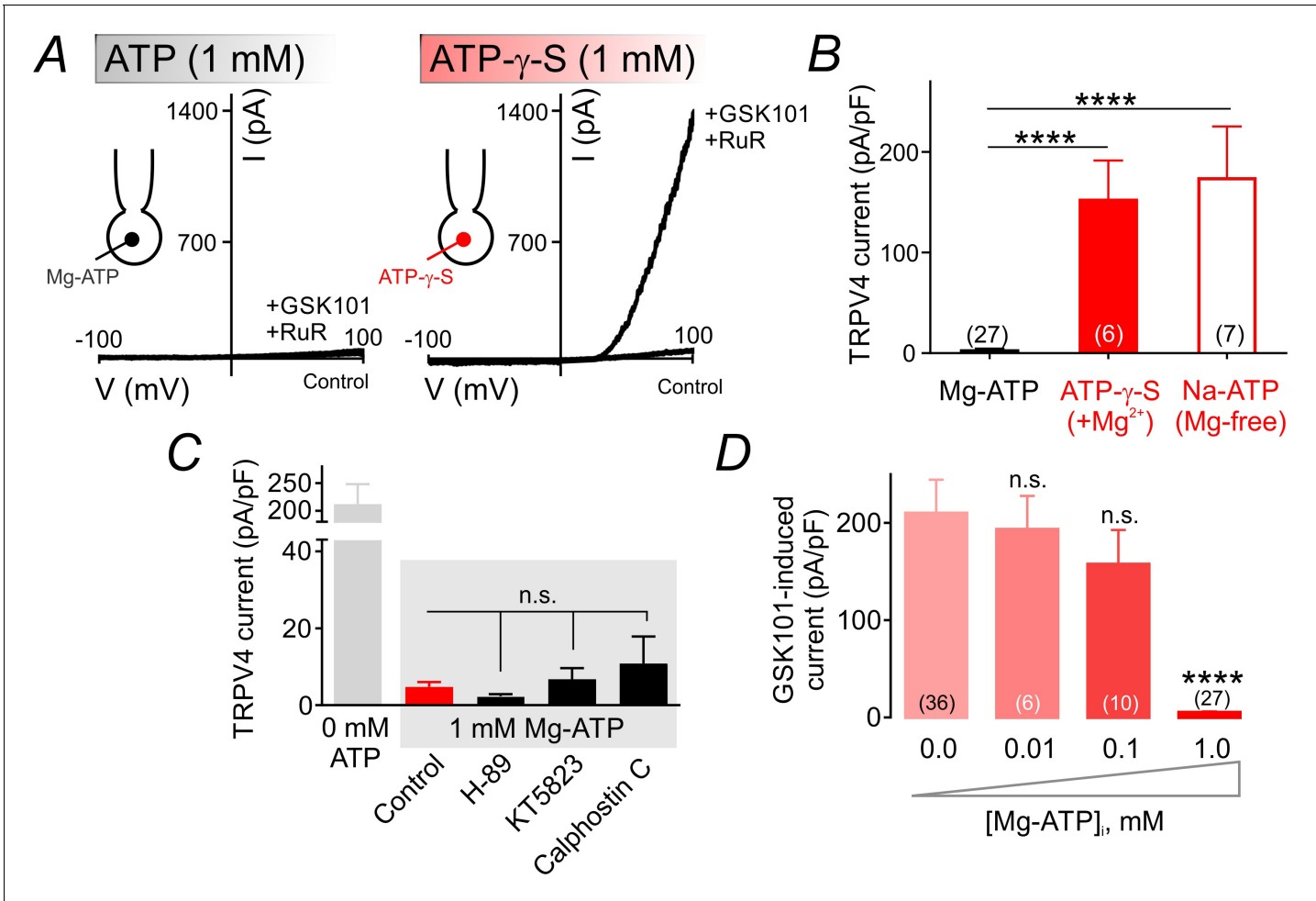

**Figure 2.** ATP hydrolysis is required for ATP-mediated suppression of TRPV4 channel activity. (**A**) Representative traces of current-voltage relationships in cECs recorded using voltage ramps (−100 to 100 mV) before and after the application of 100 nM GSK101 and 1 μM RuR. cECs were dialyzed with 1 mM Mg-ATP (*left*) or 1 mM Mg-ATP-γ-S (*right*). (**B**) Summary data showing GSK101 (100 nM)-induced outward currents at 100 mV in cECs dialyzed with Mg-ATP (1 mM), Mg-ATP-γ-S (1 mM) or Na-ATP (1 mM, in $Mg^{2+}$ free solution). A minimum duration of 5 min after the application of GSK101 was allowed for outward TRPV4 current to develop in each cEC. Data are presented as means ± SEM (****$p < 0.0001$ vs. Mg-ATP, one-way ANOVA followed by Dunnett's multiple comparisons test). (**C**) TRPV4 outward currents induced by 100 nM GSK101 at 100 mV, recorded from dialyzed cECs (0 and 1 mM Mg-ATP). Mg-ATP–dialyzed cECs (gray shadow) were pre-treated with inhibitors of PKA (H-89, 1 μM), PKG (KT5823, 1 μM) or PKC (calphostin C, 0.5 μM) for ~10–15 min prior to GSK101 application, or left untreated (control). Data are presented as means ± SEM (n.s. denotes not significant vs. control Mg-ATP, one-way ANOVA, Dunnett's multiple comparisons test, n = 6–24). (**D**) Summary data showing the effect of raising intracellular Mg-ATP concentration on GSK101-induced TRPV4 currents. Data are means ± SEM (****$p < 0.0001$, one-way ANOVA followed by Dunnett's multiple comparisons test).

DOI: https://doi.org/10.7554/eLife.38689.010

The following source data is available for figure 2:

**Source data 1.** Numerical data that were used to generate the chart in *Figure 2B*.
DOI: https://doi.org/10.7554/eLife.38689.011
**Source data 2.** Numerical data that were used to generate the chart in *Figure 2C*.
DOI: https://doi.org/10.7554/eLife.38689.012
**Source data 3.** Numerical data that were used to generate the chart in *Figure 2D*.
DOI: https://doi.org/10.7554/eLife.38689.013

kinase. Coordination of $Mg^{2+}$ is critical for kinase-mediated phosphoryl transfer reactions (*O'Rourke, 1993*; *Ryves and Harwood, 2001*; *Zakharian et al., 2011*). Thus, the prediction is that ATP would be unable to inhibit TPRV4 currents in the absence of the cofactor $Mg^{2+}$ if a kinase underlies Mg-ATP–mediated TRPV4 inhibition. Consistent with this prediction, replacing Mg-ATP in the patch pipette with the $Na^+$ salt of ATP (Na-ATP, 1 mM) while eliminating $Mg^{2+}$ from the pipette solution abolished the suppressive effect of ATP (*Figure 2B*; *Figure 2—source data 1*), suggesting the contribution of a kinase to this process.

On the basis of these observations, we next tested the involvement of protein kinases by bath-applying the protein kinase A (PKA) inhibitor H-89 (1 µM), protein kinase G (PKG) inhibitor KT5823 (1 µM), or protein kinase C (PKC) inhibitor calphostin C (0.5 µM). We then monitored outward TRPV4-mediated currents as described above. None of these inhibitors affected currents in cells dialyzed with Mg-ATP (*Figure 2C*; *Figure 2—source data 2*), ruling out a role for these protein kinases in TRPV4 inhibition. Further support for this conclusion is provided by the observation that lower concentrations of Mg-ATP (10 and 100 µM)—which activate the majority of protein kinases (*Knight and Shokat, 2005*)—failed to effectively inhibit TRPV4 currents in cECs (*Figure 2D*; *Figure 2—source data 3*).

## ATP-dependent suppression of TRPV4 channels is mediated by $PIP_2$

Lipid kinases, like protein kinases, also hydrolyze ATP, but typically require higher ATP concentrations (*Hilgemann, 1997*; *Knight and Shokat, 2005*; *Xie et al., 1999*). The requirement for millimolar Mg-ATP concentrations to effectively inhibit TRPV4 currents (*Figure 1C*, *Figure 2D*) is thus more consistent with lower ATP-affinity lipid kinases like those involved in phosphoinositide synthesis (*Balla and Balla, 2006*). In this cascade (*Figure 3A*), phosphatidylinositol 4-kinase (PI4K) utilizes ATP to phosphorylate phosphatidylinositol (PI), converting it to phosphatidylinositol 4-phosphate (PIP), which in turn is phosphorylated to $PIP_2$ by phosphatidylinositol 4-phosphate 5-kinase (PIP5K), a step that also requires ATP. Thus, the cellular levels of $PIP_2$ are maintained through an ATP-hydrolysis–dependent process (*Baukrowitz et al., 1998*; *Hilgemann, 1997*; *Suh and Hille, 2002*). To assess the possible involvement of this pathway in capillary TRPV4 suppression, we tested the effects of four structurally unrelated pharmacological inhibitors of PI4K: wortmannin (50 µM), PIK93 (300 nM), phenylarsine oxide (PAO; 30 µM), and LY294002 (300 µM). In each case, inhibitors of $PIP_2$ synthesis significantly reversed TRPV4 channel inhibition by ATP (*Figure 3B*; *Figure 3—source data 1*), despite the high intracellular Mg-ATP concentration (1 mM). Notably, at concentrations below those necessary to inhibit PI4K (but sufficient to suppress phosphoinositide 3-kinases), the inhibitors wortmannin (0.1 µM) and LY294002 (10 µM) failed to increase TRPV4 current in cECs (*Figure 3B*; *Figure 3—source data 1*). Collectively, these observations indicate the involvement of the ATP/PI4K regulatory axis, and presumably $PIP_2$, in suppressing TRPV4 channels in cECs.

$PIP_2$ is a well-established regulator of membrane proteins, including ion channels (*Hille et al., 2015*; *Huang et al., 1998*; *Nilius et al., 2008*). To directly test whether $PIP_2$ inhibits TRPV4 channels in cECs, we introduced the water-soluble, short acyl chain dioctanoyl $PIP_2$ (diC8-$PIP_2$) via the patch pipette. In the absence of Mg-ATP, diC8-$PIP_2$ inhibited GSK101-induced TRPV4 currents, reducing them by 70–80% at concentrations of 10 to 50 µM (*Figure 3C,D*; *Figure 3—source data 2*). Similar inhibition was observed with the longer-chain $PIP_2$ analog, diC16-$PIP_2$ (10 µM) (*Figure 3—figure supplement 1*). Moreover, scavenging endogenous, negatively charged $PIP_2$ with intracellular poly-L-lysine (3 µg/ml, molecular weight 15,000–30,000), included in the patch pipette, attenuated Mg-ATP effects on GSK101-induced TRPV4 currents (*Figure 3E,F*; *Figure 3—source data 3*). Taken together, these data show that $PIP_2$ inhibits both GSK101-induced and constitutive TRPV4 channel activity in cECs.

## $G_qPCR$ activation relieves TRPV4 suppression in cECs by promoting $PIP_2$ hydrolysis

The primary mechanism responsible for reducing $PIP_2$ levels is activation of $G_qPCRs$ and subsequent PLC-mediated $PIP_2$ hydrolysis. To determine whether endothelial $G_qPCR$-activation–mediated $PIP_2$ depletion relieves the inhibition of TRPV4 channels, we examined the effects of $G_qPCR$ agonists on constitutive single-channel TRPV4 currents in cECs under simulated physiological conditions (i. e. dialyzed with Mg-ATP) (*Figure 1D*). We first tested the effects of prostaglandin E2 ($PGE_2$), a

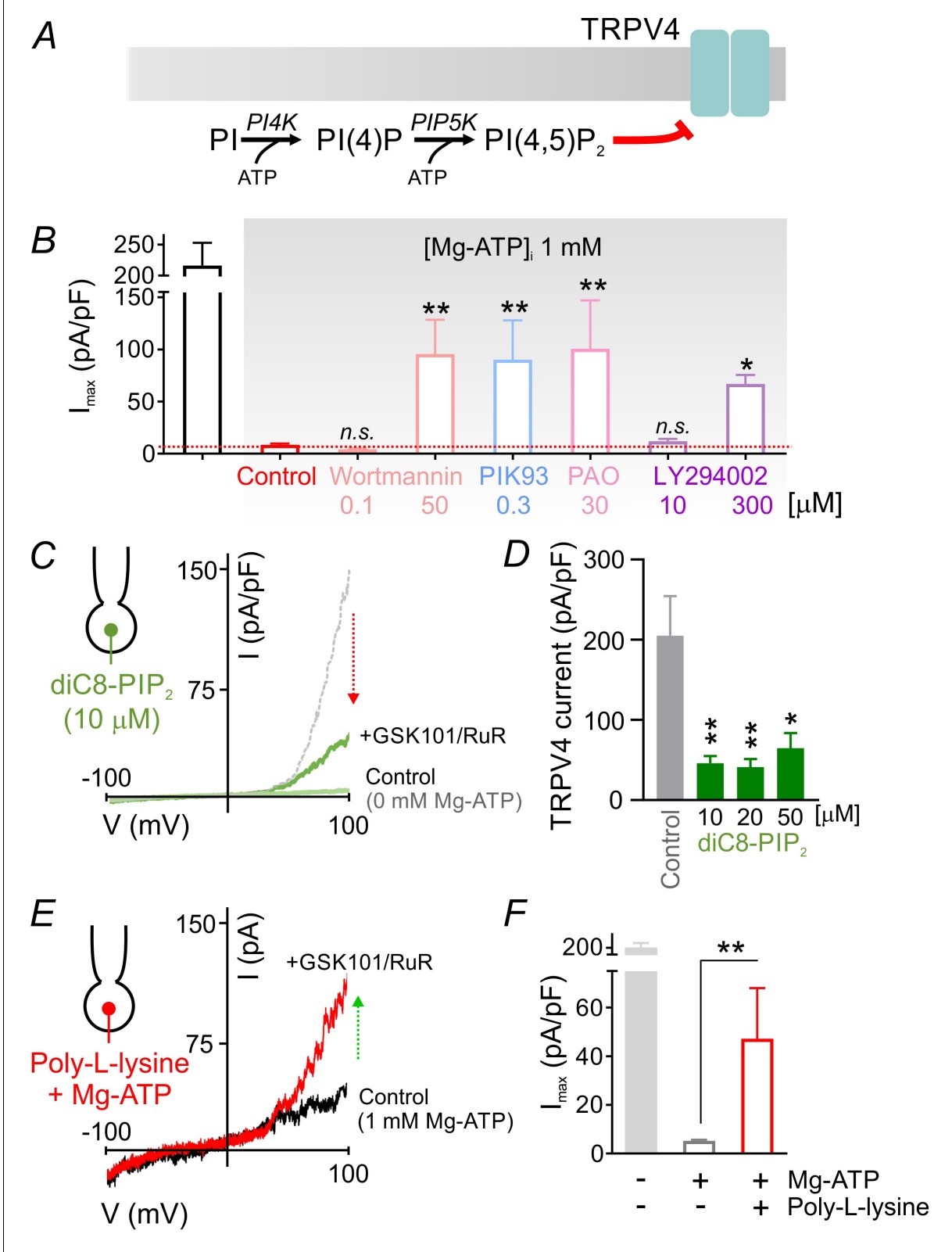

**Figure 3.** PIP$_2$ mediates tonic inhibition of capillary TRPV4 channels. (**A**) Schematic diagram showing the ATP-dependent synthesis steps leading to the production of PIP$_2$. (**B**) Average maximum outward TRPV4 current induced by 100 nM GSK101, recorded in cECs at 100 mV using the conventional whole-cell configuration. cECs dialyzed with 1 mM Mg-ATP were treated for ~10 min with wortmannin (0.1, 50 μM), PIK93 (0.3 μM), PAO (30 μM) or LY294002 (10, 300 μM), or were left untreated (control). A minimum duration of 10–15 min after the application of GSK101 was allowed for outward

*Figure 3 continued on next page*

*Figure 3 continued*

TRPV4 current to develop in each cEC. Data are means ± SEM (**p<0.01, *p<0.05 vs. control Mg-ATP, one-way ANOVA followed by Dunnett's multiple comparisons test; n = 6–27). (C) Traces of current-voltage relationship obtained from a cEC dialyzed with 10 µM diC8-PIP$_2$ and 0 mM Mg-ATP using a voltage ramp (−100 to 100 mV) before and after (green) the application of GSK101 (100 nM) and RuR (1 µM). The dotted gray trace is a representative GSK101-induced current recorded from a control cEC dialyzed with 0 µM diC8-PIP$_2$ and 0 mM Mg-ATP. (D) Summary data showing GSK101 (100 nM)-induced currents at 100 mV in cECs dialyzed with different concentrations of diC8-PIP$_2$ (10, 20, 50 µM) or 0 µM phosphoinositide (control). The pipette solution lacked Mg-ATP in all groups. GSK101-evoked outward currents developed over ~5 min. Data are presented as means ± SEM (*p<0.05, **P<0.01, one-way ANOVA followed by Dunnett's multiple comparisons test; n = 10–18). (E, F) Representative trace (E) and summary data showing GSK101-induced currents in cECs dialyzed with 1 mM Mg-ATP and poly-L-lysine (3 µg/ml). A duration of 10 min was allowed after the application of GSK101 for outward TRPV4 current to develop in each cEC. Data in F are presented as means ± SEM (**p<0.01, unpaired Student's t-test; n = 8–18).

DOI: https://doi.org/10.7554/eLife.38689.014

The following source data and figure supplements are available for figure 3:

**Source data 1.** Numerical data that were used to generate the chart in *Figure 3B*.
DOI: https://doi.org/10.7554/eLife.38689.017
**Source data 2.** Numerical data that were used to generate the chart in *Figure 3D*.
DOI: https://doi.org/10.7554/eLife.38689.018
**Source data 3.** Numerical data that were used to generate the chart in *Figure 3F*.
DOI: https://doi.org/10.7554/eLife.38689.019
**Figure supplement 1.** The long-acyl chain PIP$_2$, diC16-PIP$_2$, suppresses TRPV4 currents.
DOI: https://doi.org/10.7554/eLife.38689.015
**Figure supplement 2.** 11,12-EET-induced currents in cECs.
DOI: https://doi.org/10.7554/eLife.38689.016

postulated NVC agent that acts on G$_q$-coupled EP$_1$ receptors in cECs to deplete PIP$_2$ (*Harraz et al., 2018*) and has previously been proposed to act through G$_s$-protein-coupled EP$_2$ and EP$_4$ receptors to cause vasodilation (*Lacroix et al., 2015*; *Zonta et al., 2003*). PGE$_2$ significantly increased TRPV4 single-channel open probability in the absence of GSK101 (*Figure 4A,B*), increasing NP$_O$ ~6 fold (*Figure 4C*; *Figure 4—source data 1*). This enhancement was similar to that observed under conditions in which intracellular Mg-ATP was excluded from the pipette solution (in the absence of either a receptor agonist or GSK101; *Figure 1D*), suggesting that PGE$_2$ acts through PIP$_2$ depletion to relieve PIP$_2$-mediated inhibition and restore TRPV4 channel activity.

To confirm that PIP$_2$ depletion underlies PGE$_2$-induced activation of TRPV4 channels, we next employed a series of conventional whole-cell recordings and pharmacological interventions to test the different components in the proposed pathway. In these experiments, cECs were dialyzed with 1 mM Mg-ATP—a maneuver sufficient to significantly suppress TRPV4 channels even in the presence of 100 nM GSK101 (*Figure 1A,C*). As predicted, application of PGE$_2$ (2 µM) in the presence of GSK101 greatly increased TRPV4 currents in cECs dialyzed with Mg-ATP (*Figure 5A,B*; *Figure 5—source data 1*). Currents evoked by PGE$_2$ application were inhibited by the TRPV4 antagonist HC-067047 (1 µM) and were absent in cECs from TRPV4$^{-/-}$ animals, confirming that they are mediated by TRPV4 channels (*Figure 5C*; *Figure 5—figure supplement 1*). Introduction of the PIP$_2$ analog diC8-PIP$_2$ (10 µM) in the pipette solution or inhibition of PLC using U73122 (10 µM)—pharmacological interventions that serve to compensate for or prevent PLC-dependent PIP$_2$ degradation, respectively—prevented the increase in TRPV4 currents by PGE$_2$ (*Figure 5C*; *Figure 5—figure supplement 1*; *Figure 5—source data 2*). The non-selective prostanoid receptor (EP$_1$/EP$_2$/EP$_3$) antagonist AH6809 (10 µM) also prevented this effect of PGE$_2$, as did the selective EP$_1$ antagonist SC51322 (1 µM), suggesting that PGE$_2$ acts through the EP$_1$ receptor. U73343, the inactive analog of the PLC inhibitor U73122, did not alter PGE$_2$ effects (*Figure 5C*; *Figure 5—source data 2*). Collectively, these observations suggest that PGE$_2$ signals through the EP$_1$-PLC pathway to deplete PIP$_2$ and thereby relieve PIP$_2$-mediated TRPV4 inhibition.

The PIP$_2$ breakdown products, DAG and IP$_3$, stimulate PKC activity and promote Ca$^{2+}$ release from the endoplasmic reticulum, respectively. To rule out the involvement of these pathways in the PGE$_2$-induced increase in TRPV4 currents, we first tested the effects of the PKC inhibitors, Gö6976 (1 µM) and calphostin C (0.5 µM), and found that neither altered PGE$_2$-induced activation of TRPV4 channels (*Figure 5C*; *Figure 5—source data 2*). To determine whether intracellular Ca$^{2+}$ signals downstream of EP$_1$-PLC activation by PGE$_2$ are involved in TRPV4 disinhibition, we blocked Ca$^{2+}$

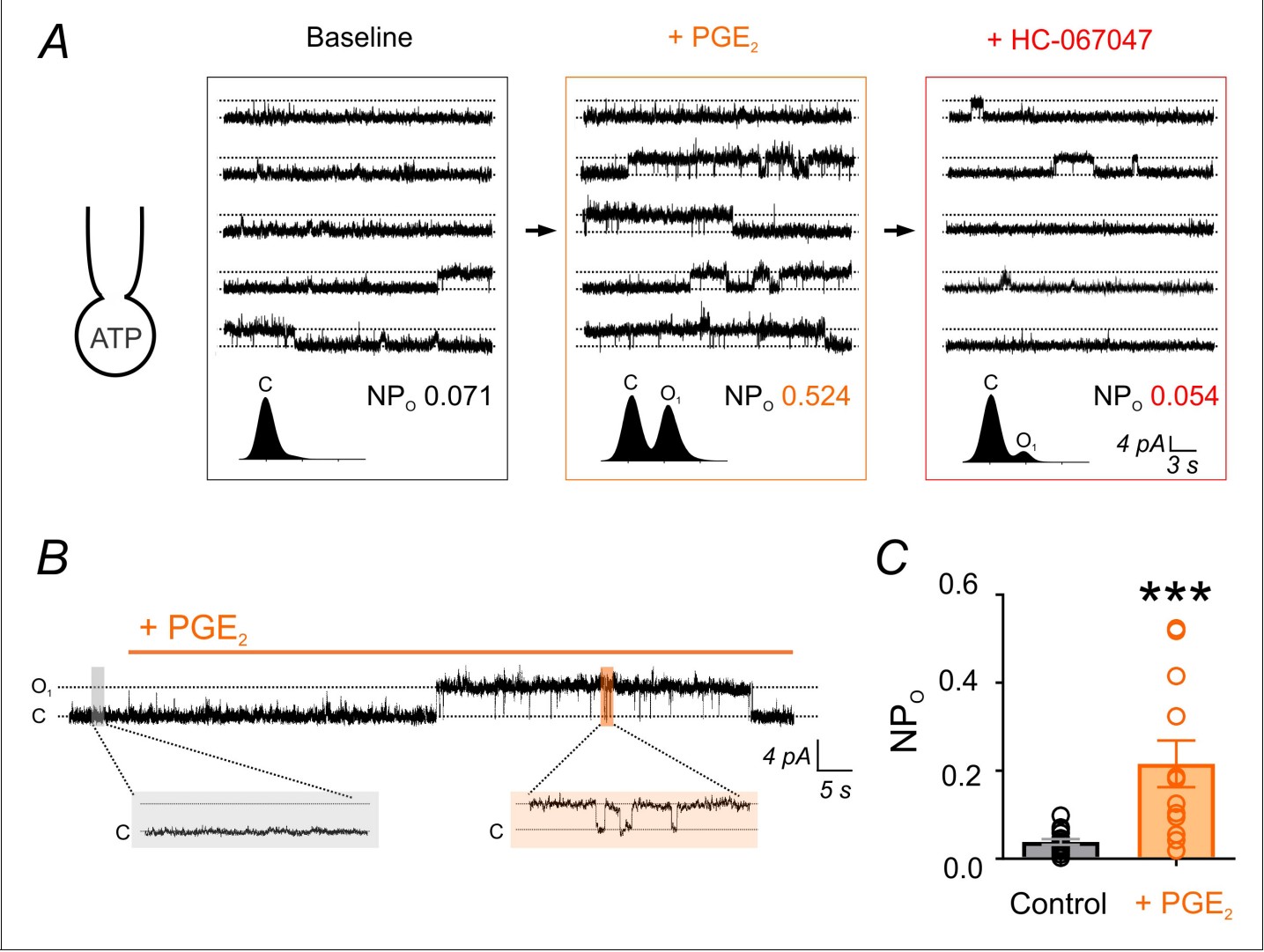

**Figure 4.** $PGE_2$ enhances TRPV4 channel activity. (**A**) *Top:* Representative conventional whole-cell recordings from a cEC dialyzed with 1 mM Mg-ATP in the absence of GSK101 and held at a membrane potential of +50 mV. Quantal outward $K^+$ currents (unitary current, 4.6 pA; sampling rate, 20 kHz; lowpass filter frequency, 1 kHz), reflecting single-channel openings, were recorded before (baseline) and after the consecutive application of $PGE_2$ (2 µM) and HC-067047 (1 µM). *Bottom:* Corresponding amplitude histograms and open probability ($NP_O$) values. (**B, C**) Representative trace (**B**) and individual-value plot (**C**) of TRPV4 $NP_O$ in cECs (dialyzed with 1 mM Mg-ATP, held at +50 mV) in the absence (control; n = 16) and presence (n = 12) of 2 µM $PGE_2$. Data in *C* are means (column bars) ± SEM (error bars, ***p<0.001, unpaired Student's t-test). Each data point represents a recording from a cEC; the average duration of each recording was 5 min.

DOI: https://doi.org/10.7554/eLife.38689.020

The following source data is available for figure 4:

**Source data 1.** Numerical data that were used to generate the chart in *Figure 4C*.

DOI: https://doi.org/10.7554/eLife.38689.021

reuptake into intracellular stores by inhibiting the sarcoplasmic/endoplasmic reticulum $Ca^{2+}$ ATPase (SERCA) pump with cyclopiazonic acid (CPA; 30 µM) or by rapidly chelating cytoplasmic $Ca^{2+}$ with BAPTA (5.4 mM). Neither maneuver attenuated $PGE_2$-induced disinhibition of TRPV4 channel activity (*Figure 5C*; *Figure 5—source data 2*), arguing against a major contribution of DAG-PKC or $IP_3$-$Ca^{2+}$ signaling to this effect. Taken together, our data show that $PGE_2$ activates $EP_1$ receptors and downstream PLC to deplete $PIP_2$ and thereby relieve TRPV4 channel inhibition, independently of $PIP_2$ metabolites.

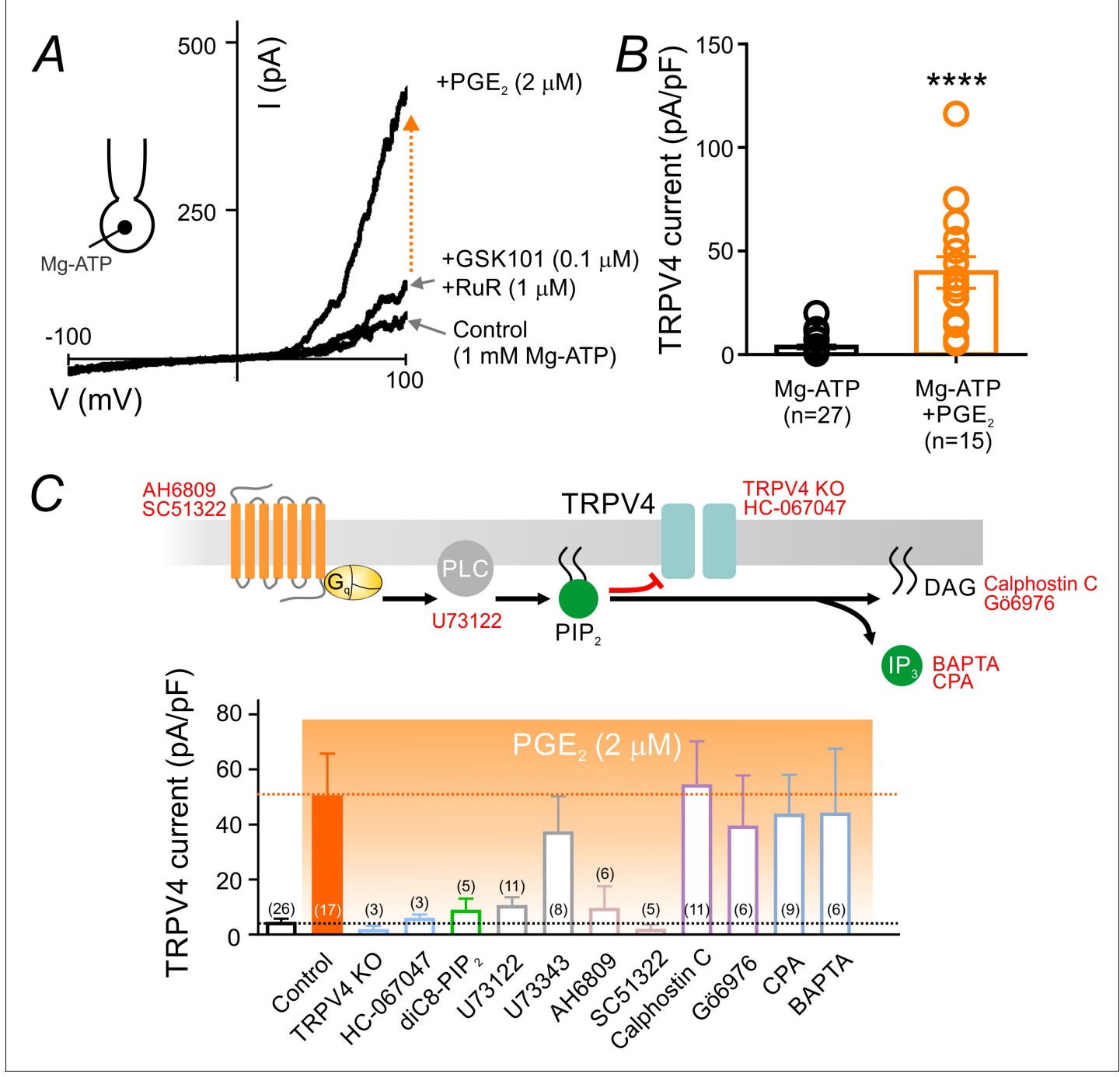

**Figure 5.** PGE$_2$ relieves PIP$_2$-mediated TRPV4 channel suppression. (**A**) Representative current-voltage plots obtained from a cEC dialyzed with 1 mM Mg-ATP and treated consecutively with GSK101 (100 nM) and RuR (1 μM) followed by 2 μM PGE$_2$. (**B**) Summary individual-value plot of GSK101-induced TRPV4 currents at 100 mV in cECs dialyzed with 1 mM Mg-ATP in the absence (black; n = 27) and presence (orange; n = 15) of 2 μM PGE$_2$. Incubation of cECs with PGE$_2$ lasted ~15 min. Data in *B* are means (column bars) ± SEM (error bars, ****p<0.0001, unpaired Student's t-test). (**C**) *Top:* Schematic diagram showing the G$_q$PCR-dependent hydrolysis of PIP$_2$ and the interventions used to test different components of the proposed pathway. *Bottom:* Summary data showing GSK101 (100 nM)-induced currents recorded at 100 mV in cECs dialyzed with 1 mM Mg-ATP. Currents were recorded in the absence and presence of 2 μM PGE$_2$ (orange shading), with or without (control) the indicated interventions. Concentrations (and application method): HC-067047, 1 μM (bath); diC8-PIP$_2$, 10 μM (pipette), U73122, 10 μM (bath); U73343, 10 μM (bath); AH6809, 10 μM (bath); SC51322, 1 μM (bath); calphostin C, 0.5 μM (bath); Gö6976, 1 μM (bath); CPA, 30 μM (bath); BAPTA, 5.4 mM (pipette). For bath application, pharmacological agents were added 10–15 min before PGE$_2$ application.

DOI: https://doi.org/10.7554/eLife.38689.022

*Figure 5 continued on next page*

*Figure 5 continued*

The following source data and figure supplements are available for figure 5:

**Source data 1.** Numerical data that were used to generate the chart in *Figure 5B*.
DOI: https://doi.org/10.7554/eLife.38689.025
**Source data 2.** Numerical data that were used to generate the chart in *Figure 5C*.
DOI: https://doi.org/10.7554/eLife.38689.026
**Figure supplement 1.** $PGE_2$ relieves TRPV4 current inhibition.
DOI: https://doi.org/10.7554/eLife.38689.023
**Figure supplement 2.** Different $G_q$PCR agonists differentially regulate TRPV4 activity.
DOI: https://doi.org/10.7554/eLife.38689.024

Carbachol (10 µM), which can signal through $G_q$-coupled muscarinic receptors, induced an increase in TRPV4 currents comparable to that produced by $PGE_2$ (*Figure 5—figure supplement 2*), indicating that $G_q$PCR-mediated TRPV4 activation is not restricted to prostanoid $EP_1$ receptors. However, neither the purinergic receptor agonist, ATP (50 µM), nor the protease-activated receptor-2 (PAR2)-activating peptide, $SLIGRL-NH_2$ (5 µM), affected TRPV4 channel activity (*Figure 5—figure supplement 2*). The reason for these apparent differential effects of various $G_q$PCRs is currently unclear, but could reflect differences in expression levels or cellular localization of the corresponding receptors, or receptor desensitization owing to ligand-dependent proteolysis (PAR2) or receptor internalization (e.g. P2Y subtypes)(*Cho et al., 2005a, 2005b*; *Dickson et al., 2013*; *Jung et al., 2016*; *Sromek and Harden, 1998*; *Vanlandewijck et al., 2018*).

## $G_q$PCR signaling as a 'switch' that shifts the balance between TRPV4 and Kir2.1 signaling

$PIP_2$ levels are key to the maintenance and activation of inward rectifier $K^+$ channels, as reported recently by our group (*Harraz et al., 2018*) and others (*D'Avanzo et al., 2010*; *Hansen et al., 2011*; *Huang et al., 1998*). Our current (*Figure 5*) and previous (*Harraz et al., 2018*) findings indicate that $G_q$PCR-activation–induced depletion of $PIP_2$ exerts opposite effects on endothelial TRPV4 (*activation*) and Kir2.1 (*inhibition*) channels. In fact, $G_q$PCR activation in cECs deactivates Kir2.1 currents through $PIP_2$ hydrolysis and cripples capillary-to-arteriolar electrical signaling (*Harraz et al., 2018*). It is thus conceivable that $G_q$PCR-mediated signaling could shift the balance between TRPV4 and Kir2.1 signaling in brain capillaries through perturbation of endothelial $PIP_2$ levels. To test this, we designed an experiment that allowed simultaneous monitoring of TRPV4 and Kir2.1 currents in the context of $G_q$PCR-mediated changes in endothelial $PIP_2$ levels. Using the perforated-patch configuration to maintain physiological levels of ATP and $PIP_2$, and bathing cECs in a solution containing an extracellular $K^+$ concentration ($[K^+]_o$) of 60 mM to facilitate monitoring of Kir2.1 currents, we evoked TRPV4 currents with 2 µM GSK101, a concentration sufficient to sub-maximally activate an outward current in this configuration (in the presence of 1 µM RuR to block inward TRPV4 current). In this setting, application of PGE (2 µM) enhanced TRPV4 currents and simultaneously inhibited Kir2.1 currents (*Figure 6A,B*; *Figure 6—source data 1*) compared with matching time controls, which showed no significant change in either current (*Figure 6—figure supplement 1*). The onset of changes in TRPV4 and Kir2.1 currents in response to $PGE_2$ was rapid (<60 s). A kinetic analysis revealed that TRPV4 activation and Kir2.1 suppression were kinetically comparable, with times for half-maximal change in activity ($t_{0.5}$) of 3.4 min for TRPV4 and 4.4 min for Kir2.1 channels, and corresponding time constants ($\tau$) of 4.9 and 6.4 min, respectively (*Figure 6C*; *Figure 6—source data 2*). The muscarinic receptor agonist carbachol (10 µM) similarly facilitated TRPV4 currents and inhibited Kir2.1 currents (*Figure 6—figure supplement 2*). In contrast, purinergic receptor stimulation with ATP did not affect TRPV4 currents (*Figure 5—figure supplement 2*; *Figure 6—figure supplement 3*) despite inhibiting Kir2.1 currents (*Figure 6—figure supplement 3*), as we reported previously (*Harraz et al., 2018*). These findings collectively establish a potential mechanism by which $PGE_2$ or other suitable $G_q$PCR agonists, could alter the balance between electrical (hyperpolarizing) signaling mediated by Kir2.1 channels and TRPV4 channel signaling in capillaries through modulation of $PIP_2$ levels (*Figure 7*).

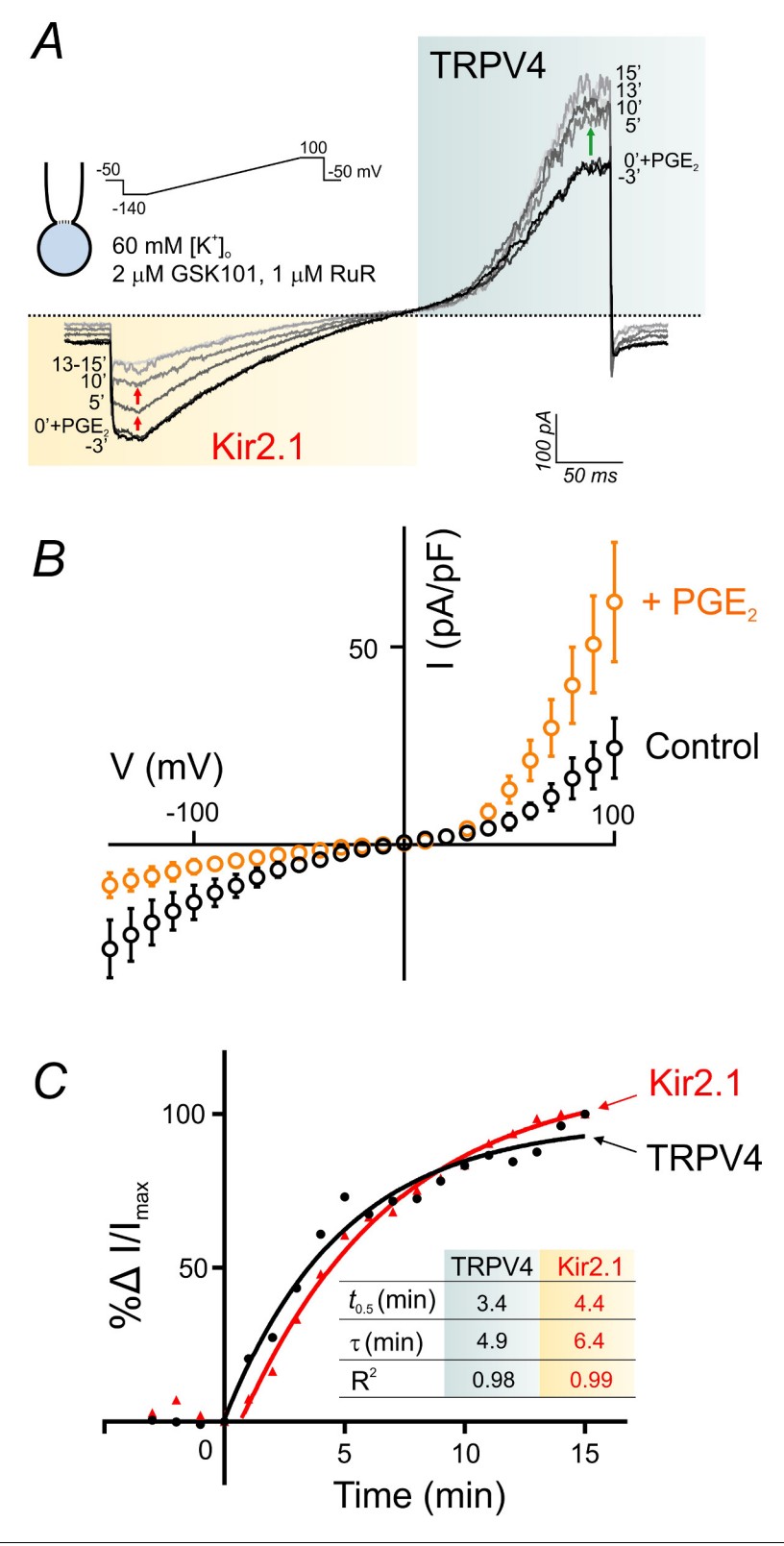

**Figure 6.** PGE$_2$ simultaneously and reciprocally regulates TRPV4 and Kir2.1 channel activities. (**A**) Representative traces illustrate simultaneous recordings of Kir2.1 (inward) and TRPV4 (outward) currents in a cEC obtained using the perforated whole-cell configuration. Voltage ramps (300 ms, −140 to 100 mV) were used and the cEC was bathed in a 60 mM [K$^+$]$_o$ solution supplemented with 2 μM GSK101 and 1 μM RuR. Traces represent currents before and for a duration of 15 min after the application of 2 μM PGE$_2$. (**B**) Averaged current-voltage relationship (n = 5 cECs) corresponding

*Figure 6 continued on next page*

*Figure 6 continued*

to the experiment in *A*, before (control) and after (maximum changes at 15 min) application of $PGE_2$. (**C**) Summary data showing the kinetics of TRPV4 current enhancement (black) and Kir2.1 current decline (red) following application of 2 µM $PGE_2$ onto cECs (as in *A*) at room temperature. Points are average percentage change in normalized currents before and over 15 min after $PGE_2$ application (n = 5). Curves are best fits of exponential change (rise: TRPV4; decay: Kir2.1). *Inset table:* kinetic parameters based on the two curve fits. Changes in currents plateaued ~13 min after the application of $PGE_2$.

DOI: https://doi.org/10.7554/eLife.38689.027

The following source data and figure supplements are available for figure 6:

**Source data 1.** Numerical data that were used to generate the chart in *Figure 6B*.

DOI: https://doi.org/10.7554/eLife.38689.031

**Source data 2.** Numerical data that were used to generate the chart in *Figure 6C*.

DOI: https://doi.org/10.7554/eLife.38689.032

**Figure supplement 1.** TRPV4 and Kir2.1 currents are preserved in cytoplasm-intact cECs.

DOI: https://doi.org/10.7554/eLife.38689.028

**Figure supplement 2.** The muscarinic receptor agonist carbachol activates TRPV4 currents and inhibits Kir2.1 currents.

DOI: https://doi.org/10.7554/eLife.38689.029

**Figure supplement 3.** Purinergic receptor activation affects Kir2.1 channels but not TRPV4 channels.

DOI: https://doi.org/10.7554/eLife.38689.030

## Discussion

We previously demonstrated that increases in $[K^+]_o$ associated with neuronal activity alter cerebral blood flow at the capillary level, showing that extracellular $K^+$ activates capillary Kir2.1 channels, triggering a retrograde electrical (hyperpolarizing) signal that propagates upstream to dilate feeding arterioles and enhance blood flow to the active region (*Longden et al., 2017*). We have also shown that $G_qPCR$-mediated $PIP_2$ depletion inhibits capillary electrical signaling (*Harraz et al., 2018*), identifying a point of intersection between electrical signaling and endothelial $G_qPCR$ activity. In the present study, we show that cECs express TRPV4 channels, and demonstrate that these channels are tonically suppressed by basal levels of $PIP_2$. Furthermore, we show that $G_qPCR$ activation relieves TRPV4 channel inhibition through $PIP_2$ depletion. This paradigm introduces the phosphoinositide $PIP_2$ as a master regulator of TRPV4 and Kir2.1 signaling in the capillary endothelium and highlights the ability of $G_qPCR$ activity to tune the balance of these signaling modalities to favor TRPV4 signaling (*Figure 7*).

The repertoire of functional ion channels in brain cECs remains incompletely characterized, although certain molecular features have recently come into focus. For instance, our evidence suggests that the inward rectifier Kir2.1 channel is the major $K^+$ channel type in brain cECs (*Longden et al., 2017*), whereas $Ca^{2+}$-sensitive SK and IK channels, which are present in arterial and arteriolar ECs and play a prominent role in regulating vascular tone (*Ledoux et al., 2008*; *Sonkusare et al., 2012*; *Taylor et al., 2003*), are not expressed in brain cECs (*Longden et al., 2017*). Notably, the identity of depolarizing ($Na^+$ and/or $Ca^{2+}$-permeable) channels in cECs, which we predict must be present to allow the membrane potential to reset to support repeated operation of our previously reported Kir2.1-dependent electrical signaling-based NVC mechanism, is not known. Our demonstration that the highly selective TRPV4 agonist, GSK101, induced currents in brain cECs that were eliminated by the TRPV4-specific antagonist, HC-067047, and were absent in cECs from TRPV4$^{-/-}$ mice firmly establishes the presence of this non-selective cation channel in capillaries. In arterial and arteriolar ECs, TRPV4 channel-mediated $Ca^{2+}$ influx is closely linked to activation of $Ca^{2+}$-sensitive SK and IK channels and subsequent membrane potential hyperpolarization (*Sonkusare et al., 2012*). However, because cECs lack functional $Ca^{2+}$-activated $K^+$ channels (*Longden et al., 2017*), TRPV4-mediated influx of $Na^+/Ca^{2+}$ in these cells instead would lead to membrane potential depolarization (*Behringer et al., 2017*; *Earley and Brayden, 2015*). Collectively, these observations suggest that the TRPV4 channel is a major depolarizing element in cECs, although we cannot rule out the possibility that other, as yet unidentified, cation channels may contribute to membrane depolarization, as suggested by earlier studies (*Csanády and Adam-Vizi, 2004*; *Popp and Gögelein, 1992*; *Csanády and Adam-Vizi, 2003*). Our observations also highlight the fact that the functional role of TRPV4 channels is critically dependent on the expression and function of key associated proteins.

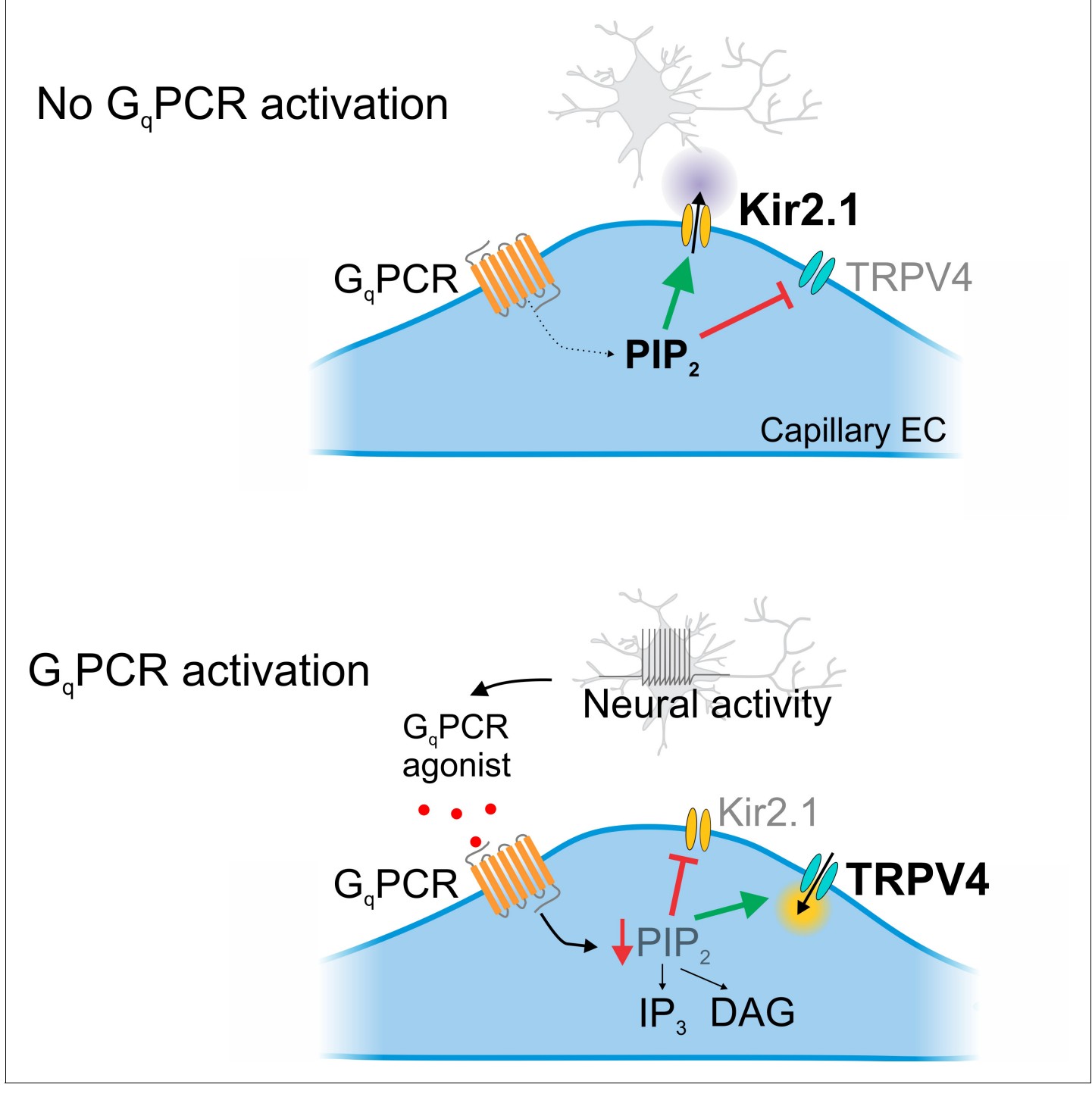

**Figure 7.** Cartoon representation of G$_q$PCR-mediated reciprocal effects on capillary ion channel activity. Schematic diagram summarizing the proposed mechanism. *Top*: In the absence of G$_q$PCR stimulation, endogenous PIP$_2$ levels are sufficient to tonically inhibit TRPV4 channels and maintain Kir2.1 channel activity. *Bottom*: G$_q$PCR activation with an agonist stimulates PIP$_2$ hydrolysis, resulting in the loss of PIP$_2$-mediated maintenance of Kir2.1 activity and inhibition of TRPV4 activity.

DOI: https://doi.org/10.7554/eLife.38689.033

Intriguingly, and in striking contrast to the case in peripheral arterial ECs (*Sonkusare et al., 2014*; *Sonkusare et al., 2012*), the open probability of capillary TRPV4 channels in cECs was remarkably low under basal conditions and was increased by dialyzing out intracellular contents—the first link in the chain leading to our discovery that TRPV4 channels in cECs are intrinsically inhibited by intracellular ATP. It is well established that lipid kinases involved in $PIP_2$ synthesis require millimolar ATP for their activity (*Balla and Balla, 2006*; *Hilgemann, 1997*; *Knight and Shokat, 2005*; *Suer et al., 2001*), and it has been amply demonstrated that $PIP_2$ maintenance is dependent on ATP in multiple cell types (*Suh and Hille, 2008*; *Ye et al., 2018*; *Zakharian et al., 2011*), including cECs (*Harraz et al., 2018*). Three major lines of evidence presented here support the conclusion that cytosolic ATP-dependent maintenance of sustained, basal levels of $PIP_2$ suppresses TRPV4 channel activity. First, millimolar concentrations of hydrolyzable ATP suppressed capillary TRPV4 channel activity. Second, inclusion of $PIP_2$ analogs in the patch pipette inhibited TRPV4 currents. Third, scavenging $PIP_2$ or inhibiting its synthesis abrogated ATP-mediated inhibition. Notably, $PIP_2$ has been reported to directly interact with different residues on the TRPV4 N-terminus in heterologous expression systems (*Garcia-Elias et al., 2013*; *Takahashi et al., 2014*). However, these studies reached discrepant conclusions, with one suggesting that direct binding of $PIP_2$ to the ankyrin repeat domain of the TRPV4 channel inhibits different modes of TRPV4 activation (*Takahashi et al., 2014*) and the other reporting that $PIP_2$ is necessary for heat-, hypotonicity- or epoxyeicosatrienoic acid (EET)-induced channel activation by facilitating structural rearrangements of the channel (*Garcia-Elias et al., 2013*). These discrepancies may reflect divergent effects of $PIP_2$ on channel behavior through binding to multiple sites in the channel. Intriguingly, the putative endogenous TRPV4 channel activator (*Earley and Brayden, 2015*; *Watanabe et al., 2003*) 11,12-epoxyeicosatrienoic acid (11,12-EET, 1 µM) evoked currents in cECs even in the absence of dialyzed $PIP_2$ (*Figure 3—figure supplement 2*). In any case, our results are the first to report $PIP_2$-mediated suppression of TRPV4 channels in the endothelium and more broadly in native cells and are congruent with the results of *Takahashi et al., 2014*. Interestingly, a recent study provided structural evidence that lipid molecules are tightly bound to the selectivity filter of the TRPV4 channel pore, although the identity and function of these lipids were not characterized (*Deng et al., 2018*).

Canonical $G_qPCR$ signaling involves PLC activation and subsequent $PIP_2$ breakdown into DAG and $IP_3$. Receptor agonists that activate $G_qPCRs$ can dramatically lower $PIP_2$ by promoting its hydrolysis at rates that exceed those of $PIP_2$ re-synthesis. We show here that $PGE_2$, which has previously been postulated to act as an NVC agent through actions on $G_s$-coupled $EP_2/EP_4$ receptors in arteriolar smooth muscle (*Lacroix et al., 2015*; *Zonta et al., 2003*), activates TRPV4 channels in cECs independent of the action of the $PIP_2$ metabolites, DAG and $IP_3$, by relieving $PIP_2$-mediated suppression (*Figure 4*, *Figure 5*). In contrast, we recently showed that $G_qPCR$ agonists exert the opposite effect on capillary Kir2.1 channels, inhibiting their ability to mediate capillary-to-arteriole electrical signaling (*Harraz et al., 2018*). In keeping with these previous findings and our current observations, simultaneous monitoring of TRPV4 and Kir2.1 channel currents confirmed the bidirectional effects of $G_qPCR$ agonists on the two channels (*Figure 6*). This two-way modulation (*Figure 7*) is unique as it indicates that a single $G_qPCR$ signaling cascade is capable of altering the balance between electrical Kir2.1 signaling (*inhibition*) and TRPV4 signaling (*facilitation*). Given the absence of $Ca^{2+}$-activated $K^+$ channels in cECs, noted above, $G_qPCR$ signaling-induced $PIP_2$ depletion would likely depolarize cECs through the simultaneous disinhibition of depolarizing TRPV4 channels (this study) and deactivation of hyperpolarizing Kir2.1 channels (*Harraz et al., 2018*).

Based on our direct Kir2.1 current measurements (*Harraz et al., 2018*; *Longden et al., 2017*) and the known voltage- and $K^+$-dependence of Kir2.1 channels (*Longden and Nelson, 2015*), we estimate that the outward current through these channels at physiological membrane potentials (about $-40$ mV) and external $K^+$ (3 mM) is ~6 fA. Elevation of external $K^+$ to 10 mM would increase Kir2 current at this voltage to ~260 fA. Though seemingly miniscule, such small membrane currents are precisely what is needed to ensure conduction fidelity. We have measured a sustained and profound hyperpolarization of about $-25$ mV in arteriolar smooth muscle in response to capillary stimulation with 10 mM $K^+$ (*Longden et al., 2017*). Our computational modeling indicates that a stable membrane hyperpolarization of 25 mV requires the outward (hyperpolarizing) current to greatly exceed the inward (depolarizing) current. Our estimate of basal, $PIP_2$-suppressed TRPV4 current at $-40$ mV is about $-80$ fA. $G_qPCR$ activation increases open probability ~6 fold, producing a current (400–500 fA) sufficient to effectively short circuit $K^+$-induced hyperpolarization and cripple this key

Kir2.1-based NVC mechanism. In conclusion, the low level of TRPV4 channel activity aligns with the functional role of capillaries.

Activation of capillary $G_qPCRs$ should also trigger $Ca^{2+}$ signals in cECs, presumably through $IP_3$-mediated $Ca^{2+}$ release from intracellular stores as well as disinhibition of TRPV4 channels (as described here) and subsequent $Ca^{2+}$ influx. Such capillary $Ca^{2+}$ signals should positively influence hemodynamics—and thus functional hyperemia—providing a plausible explanation for the role of $G_qPCR$ agonists in the NVC process, presumably through the $Ca^{2+}$-dependent activation of nitric oxide synthase (NOS) and subsequent release of the vasodilator, NO (*Förstermann et al., 1991*; *Marziano et al., 2017*). Capillary $Ca^{2+}$ signaling might also serve an entirely different purpose in functional hyperemia: because local $Ca^{2+}$ signals represent sites of $PIP_2$ depletion (through $G_qPCR$ signaling), and thus membrane potential depolarization (through TRPV4 disinhibition and Kir2.1 deactivation), they are likely to interfere with the progression of electrical signals generated further down the vascular tree. Accordingly, these signals may represent 'stop signs' or 'speed bumps' that play a role in redirecting hyperpolarizing (vasodilatory) signals away from certain brain areas, in addition to their role in resetting the capillary membrane potential, noted above.

Given the role of $PIP_2$ in negatively regulating TRPV4 channels, the exceedingly low basal TRPV4 activity and diminished sensitivity to GSK101 in cECs (*Figure 1*) compared with mesenteric artery ECs (*Sonkusare et al., 2012*) suggest differences in the $PIP_2$ 'set point' in these two vascular beds. This supposition has important physiological implications for electrical signaling in the brain. Our data suggest that $PIP_2$ levels in brain cECs are sufficient to saturate Kir2.1 channels (*Harraz et al., 2018*) and tune TRPV4 channel currents to levels that prevent inappropriate depolarization of the membrane potential (*Figure 3*). In the absence of $PIP_2$ suppression of TRPV4 channel-mediated inward currents, it is unlikely that increased outward $K^+$ currents through Kir2.1 channels in response to elevated $K^+$ would be sufficient to cause membrane potential hyperpolarization. The mechanistic basis for the apparently higher $PIP_2$ set point in brain cECs is currently unknown, but could include lower constitutive $G_qPCR$ activity, lower microenvironmental levels of $G_qPCR$ agonists and/or decreased $G_qPCR$ expression—and thus diminished $PIP_2$ breakdown. Along these lines, a recent single-cell transcriptomics analysis showed that $G\alpha_q$ (*Gnaq*) transcript levels in the mouse brain endothelium are reduced compared with those in peripheral pulmonary ECs (*Vanlandewijck et al., 2018*). Alternatively, differences in the set point could be indicative of more robust $PIP_2$ synthesis in brain capillaries, a speculation that is supported by the higher mitochondrial content—and hence ATP synthesis—in highly active brain cECs compared with other ECs (*Oldendorf and Brown, 1975*; *Oldendorf et al., 1977*).

In conclusion, this study provides compelling evidence that brain capillaries express TRPV4 channels and further shows that the activity of these channels is physiologically suppressed by basal levels of $PIP_2$. This introduces $PIP_2$ and its modulation by $G_qPCR$ agonists as major regulators of brain capillary signaling. When maintained at sufficient levels, $PIP_2$ inhibits TRPV4 channels and supports capillary-to-arteriole electrical signaling, but in response to $G_qPCR$ activation, $PIP_2$ levels are reduced, enhancing TRPV4 signaling and inhibiting retrograde electrical signaling via Kir2.1 channels. Capillary $G_qPCRs$ can therefore be envisioned as molecular switches that dictate the signaling modality in the brain microvasculature.

# Materials and methods

**Key resources table**

| Reagent type (species) or resource | Designation | Source or reference | Identifiers | A |
|---|---|---|---|---|
| Strain, strain background (*Mus musculus*, males) | TRPV4 Knockout (TRPV4$^{-/-}$) mice, C57BL/6J background | *Thorneloe et al. (2008)* | PMID: 18499743 | |
| Strain, strain background (*Mus musculus*, males) | C57BL/6J | The Jackson Laboratory | RRID:IMSR_JAX:000664 | |
| Chemical compound, drug | GSK1016790A (GSK101) | Sigma | Cat#: G0798 | |

*Continued on next page*

*Continued*

| Reagent type (species) or resource | Designation | Source or reference | Identifiers | A |
|---|---|---|---|---|
| Chemical compound, drug | Ruthenium red (RuR) | Sigma | Cat#: R2751 | |
| Chemical compound, drug | Adenosine 5'-triphosphate magnesium salt (Mg-ATP) | Sigma | Cat#: A9187 | |
| Chemical compound, drug | Adenosine 5'-triphosphate sodium salt | Sigma | Cat#: A2383 | |
| Chemical compound, drug | HC-067047 | Sigma | Cat#: SML0143 | |
| Chemical compound, drug | KT5823 | Sigma | Cat#: K1388 | |
| Chemical compound, drug | PIK93 | Tocris | Cat#: 6440 | |
| Chemical compound, drug | Phenylarsine oxide (PAO) | Sigma | Cat# P3075 | |
| Chemical compound, drug | H-89 dihydrochloride | Tocris | Cat# 2910 | |
| Chemical compound, drug | Calphostin C | Tocris | Cat#: 1626 | |
| Chemical compound, drug | LY294002 hydrochloride | Tocris | Cat#: 1130 | |
| Chemical compound, drug | AH6809 | Tocris | Cat#: 0671 | |
| Chemical compound, drug | SC51322 | Tocris | Cat#: 2791 | |
| Chemical compound, drug | U-73122 | Sigma | Cat#: U6756 | |
| Chemical compound, drug | U-73343 | Sigma | Cat#: U6881 | |
| Chemical compound, drug | Poly-L-lysine hydrochloride | Sigma | Cat# 2658 | |
| Chemical compound, drug | Gö6976 | Calbiochem | Cat#: 365250 | |
| Chemical compound, drug | Wortmannin | Sigma | Cat#: W1628 | |
| Chemical compound, drug | PI(4,5)P2 (1,2-dioctanoyl) (sodium salt) | Cayman | Cat#: 64910 | |
| Chemical compound, drug | PI(4,5)P2 (1,2-dipalmitoyl) (sodium salt) | Cayman | Cat#: 10008115 | |
| Chemical compound, drug | Prostaglandin $E_2$ | Sigma | Cat#: 5640 | |
| Chemical compound, drug | SLIGRL-NH2 | Tocris | Cat#: 1468 | |
| Chemical compound, drug | Carbachol | Sigma | Cat#: C4382 | |
| Chemical compound, drug | Cyclopiazonic acid from *Penicillium cyclopium* (CPA) | Sigma | Cat#: C1530 | |
| Chemical compound, drug | Guanosine 5'-triphosphate sodium salt (GTP) | Sigma | Cat#: G8877 | |

*Continued on next page*

*Continued*

| Reagent type (species) or resource | Designation | Source or reference | Identifiers | A |
|---|---|---|---|---|
| Chemical compound, drug | Adenosine 5′-[γ-thio]triphosphate tetralithium salt | Sigma | Cat#: A1388 | |
| Chemical compound, drug | 11,12-Epoxyeico satrienoic acid (11,12-EET) | Sigma | Cat#: E5641 | |
| Chemical compound, drug | 4α-Phorbol 12,13-didecanoate (4α-PDD) | Sigma | Cat#: P8014 | |
| Chemical compound, drug | 1,2-Bis (2-aminophenoxy) ethane-N,N,N′,N′-tetraacetic acid tetrapotassium salt (BAPTA) | Sigma | Cat#: A9801 | |
| Software, algorithm | Prism | GraphPad | RRID:SCR_002798 https://www.graphpad.com/scientific-software/prism/ | |
| Software, algorithm | Clampfit 10.7 | Axon Instruments | RRID:SCR_011323 https://www.moleculardevices.com/products/axon-patch-clamp-system | |

## Animals

All procedures involving animals received prior approval from the University of Vermont Institutional Animal Care and Use Committee. Adult (2–3 month old) male C57BL/6J mice (Jackson Laboratories, USA) and TRPV4$^{-/-}$ mice (*Thorneloe et al., 2008*) were group-housed on a 12 hr light:dark cycle with environmental enrichment and free access to food and water. Animals were euthanized by intraperitoneal injection of sodium pentobarbital (100 mg/kg) followed by rapid decapitation.

## Chemicals

1,2-Dioctanoyl phosphatidylinositol 4,5-bisphosphate sodium salt (diC8-PIP$_2$) and 1,2-dipalmitoyl phosphatidylinositol 4,5-bisphosphate sodium salt (diC16-PIP$_2$) were purchased from Cayman Chemical (USA). 12-(2-Cyanoethyl)−6,7,12,13-tetrahydro-13-methyl-5-oxo-5H-indolo(2,3-a)pyrrolo(3,4 c)-carbazole (Gö6976) was from Calbiochem (USA). N-((1S)−1-{[4-((2S)−2-{[(2,4-Dichlorophenyl) sulfonyl] amino}−3-hydroxy-propanoyl)−1-piperazinyl]carbonyl}−3-methylbutyl)−1-benzothiophene-2-carboxamide (GSK101) and 2-Methyl-1-[3-(4-morpholinyl)propyl]−5-phenyl-N-[3-(trifluoromethyl) phenyl]−1H–pyrrole-3-carboxamide (HC-067047) were obtained from Sigma-Aldrich (USA). 6-Isopropoxy-9-xanthone-2-carboxylic acid (AH6809), 8-chloro-2-[3-[(2-furanylmethyl)thio]−1-oxopropyl]-dibenz(Z)[b,f][1,4]oxazepine-10(11H)-carboxylic acid hydrazide (SC51322), N-[2-[[3-(4-bromophenyl)−2-propenyl]amino]ethyl]−5-isoquinolinesulfonamide dihydrochloride (H-89), (1R)−2-[12-[(2R)−2-(benzoyloxy)propyl]−3,10-dihydro-4,9-dihydroxy-2,6,7,11-tetramethoxy-3,10-dioxo-1-perylenyl]−1-methylethylcarbonic acid 4-hydroxyphenyl ester (calphostin C), and 2-(4-morpholinyl)−8-phenyl-4H-1-benzopyran-4-one hydrochloride (LY 294002) were purchased from Tocris (USA). Unless otherwise noted, all other chemicals were obtained from Sigma-Aldrich.

## Capillary endothelial cell isolation

Single ECs and capillary fragments were obtained from mouse brains by mechanical disruption of two 160 µm–thick brain slices using a Dounce homogenizer. Slices were homogenized in ice-cold artificial cerebrospinal fluid (124 mM NaCl, 3 mM KCl, 2 mM CaCl$_2$, 2 mM MgCl$_2$, 1.25 mM NaH$_2$PO$_4$, 26 mM NaHCO$_3$, 4 mM glucose), and debris was removed by passing the homogenate through a 62 µm nylon mesh. Retained capillary fragments were eluted into dissociation solution composed of 55 mM NaCl, 80 mM Na-glutamate, 5.6 mM KCl, 2 mM MgCl$_2$, 4 mM glucose and 10 mM HEPES (pH 7.3), containing neutral protease (0.5 mg/mL) and elastase (0.5 mg/mL) (Worthington, USA) plus 100 µM CaCl$_2$, and incubated for 24 min at 37°C. Thereafter, 0.5 mg/ml collagenase type I (Worthington) was added and the sample was incubated for an additional 2 min at 37°C. The cell suspension was filtered and the residue was washed to remove enzymes. Single cells and small

capillary fragments were dispersed by triturating 4–6 times with a fire-polished glass Pasteur pipette. Cells were stored in ice-cold isolation medium for use the same day (within ~6 hr).

## Electrophysiology

All patch-clamp electrophysiological recordings were performed at room temperature (~22°C) in either the conventional or perforated whole-cell configuration. Currents were amplified using an Axopatch 200B amplifier, filtered at 1 kHz, digitized at 10 kHz, and stored on a computer for offline analysis with Clampfit 10.5 software. Recording pipettes were fabricated by pulling borosilicate glass (1.5 mm outer diameter, 1.17 mm inner diameter; Sutter Instruments, USA) using a Narishige puller. Pipettes were fire-polished to reach a tip resistance of ~4–6 M$\Omega$. The bath solution consisted of 80 or 134 mM NaCl, 60 or 6 mM KCl, 1 mM $MgCl_2$, 10 mM HEPES, 4 mM glucose, and 2 mM $CaCl_2$ (pH 7.4). For the conventional whole-cell configuration, pipettes were backfilled with a solution consisting of 10 mM NaOH, 11.4 mM KOH, 128.6 mM KCl, 1.1 mM $MgCl_2$, 2.2 mM $CaCl_2$, 5 mM EGTA, and 10 mM HEPES (pH 7.2). A subset of experiments utilized a $Mg^{2+}$-free solution, obtained by excluding $MgCl_2$ from the pipette solution. In another group of cells, EGTA was replaced with BAPTA (5.4 mM). For perforated-patch electrophysiology, the pipette solution was composed of 10 mM NaCl, 26.6 mM KCl, 110 mM $K^+$ aspartate, 1 mM $MgCl_2$ and 10 mM HEPES; amphotericin B (200–250 µg/ml) was freshly added on the day of the experiment. Whole-cell capacitance, measured using the cancellation circuitry in the voltage-clamp amplifier, averaged 8.6 pF.

## Calculation of TRPV4 channel numbers

TRPV4 membrane currents were measured using the conventional whole-cell configuration and physiological concentrations of salts. Ruthenium red (RuR; 1 µM) was used to block $Ca^{2+}$ influx and thereby prevent $Ca^{2+}$ overload. We have shown previously that RuR causes a voltage-dependent block of TRPV channels and is rapidly driven out of the pore by membrane depolarization; at +100 mV, RuR (1 µM) blocks approximately 19% of the TRPV4 current (*Sonkusare et al., 2012*). The number of TRPV4 channels activated by 100 nM GSK101 was estimated by dividing the GSK101-induced current at +100 mV by the unitary current at this voltage (i.e. 10 pA) (*Loukin et al., 2010*; *Watanabe et al., 2002*) and correcting for the degree of RuR block.

## Data analysis

Data are expressed as means ± standard error of the mean (SEM). Effects of a given condition/treatment on whole-cell current were compared using paired or unpaired *t* tests or analysis of variance (ANOVA), as appropriate, using Graphpad Prism 7.01 software. p values$\leq$0.05 were considered statistically significant. Additional patch-clamp data analyses were performed using Clampfit 10.5 software.

## Acknowledgement

This study was supported by a postdoctoral fellowship (17POST33650030 to OFH) and a scientist development grant (17SDG33670237 to TAL) from the American Heart Association, and grants from the Totman Medical Research Trust (to MTN), Fondation Leducq (to MTN), European Union's Horizon 2020 research and innovation programme (grant agreement No 666881, SVDs@target, to MTN), and National Institutes of Health (4P20 GM103644/4-5 to the Vermont Center on Behavior and Health (TAL); P30-GM-103498 to the COBRE imaging facility at UVM College of Medicine; and P01-HL-095488, R01-HL-121706, R37-DK-053832, 7UM-HL-1207704 and R01-HL-131181 to MTN).

## Additional information

### Funding

| Funder | Grant reference number | Author |
|---|---|---|
| American Heart Association | Postdoctoral fellowship 17POST33650030 | Osama F Harraz |
| American Heart Association | Scientist Development Grant 17SDG33670237 | Thomas A Longden |

| National Institutes of Health | P01-HL-095488 | Mark T Nelson |
|---|---|---|
| National Institutes of Health | 4P20 GM103644/4-5 to the Vermont Center on Behavior and Health | Thomas A Longden |
| The Totman Medical Research Trust | | Mark T Nelson |
| Fondation Leducq | | Mark T Nelson |
| European Union's Horizon 2020 Research and Innovation Programme | Grant agreement No 666881 SVDs@target | Mark T Nelson |
| National Institutes of Health | P30-GM-103498 to the COBRE imaging facility at UVM College of Medicine | Mark T Nelson |
| National Institutes of Health | R01-HL-121706 | Mark T Nelson |
| National Institutes of Health | R37-DK-053832 | Mark T Nelson |
| National Institutes of Health | 7UM-HL-1207704 | Mark T Nelson |
| National Institutes of Health | R01-HL-131181 | Mark T Nelson |

The funders had no role in study design, data collection and interpretation, or the decision to submit the work for publication.

## Author contributions

Osama F Harraz, Conceptualization, Data curation, Formal analysis, Funding acquisition, Investigation, Methodology, Writing—original draft, Writing—review and editing; Thomas A Longden, Funding acquisition, Investigation, Methodology, Writing—review and editing; David Hill-Eubanks, Writing—original draft, Writing—review and editing; Mark T Nelson, Conceptualization, Resources, Supervision, Funding acquisition, Writing—original draft, Writing—review and editing

## Author ORCIDs

Osama F Harraz http://orcid.org/0000-0003-2061-1100
Mark T Nelson http://orcid.org/0000-0002-6608-8784

## Ethics

Animal experimentation: This study was performed in strict accordance with the recommendations in the Guide for the Care and Use of Laboratory Animals of the National Institutes of Health. All procedures involving animals received prior approval from the University of Vermont Institutional Animal Care and Use Committee (IACUC, protocol # 14-063 and 16-010). Animals were euthanized by intraperitoneal injection of sodium pentobarbital (100 mg/kg) followed by rapid decapitation. Every effort was made to minimize suffering of animal subjects.

## Decision letter and Author response

Decision letter https://doi.org/10.7554/eLife.38689.040
Author response https://doi.org/10.7554/eLife.38689.041

# Additional files

## Supplementary files

• Transparent reporting form
DOI: https://doi.org/10.7554/eLife.38689.034

## Data availability

All data generated or analyzed during this study are included in the manuscript and supporting source data files.

The following previously published datasets were used:

| Author(s) | Year | Dataset title | Dataset URL | Database, license, and accessibility information |
|---|---|---|---|---|
| Vanlandewijck M, He L, Mäe M, Andrae J, Betsholtz C | 2017 | Single cell RNA-seq of mouse brain vascular transcriptomes | https://www.ncbi.nlm.nih.gov/geo/query/acc.cgi?acc=GSE98816 | Publicly available at the NCBI Gene Expression Omnibus (accession no. GSE98816) |
| Vanlandewijck M, He L, Mäe M, Andrae J, Betsholtz C | 2017 | Single cell RNA-seq of mouse lung vascular transcriptomes | https://www.ncbi.nlm.nih.gov/geo/query/acc.cgi?acc=GSE99235 | Publicly available at the NCBI Gene Expression Omnibus (accession no. GSE99235) |

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
