## [Decision Letter]

Thank you for submitting your article "PIP_2_ depletion promotes TRPV4 channel activity in brain capillary endothelial cells" for consideration by *eLife*. Your article has been reviewed by three peer reviewers, including László Csanády as the Reviewing Editor and Reviewer #1, and the evaluation has been overseen by Richard Aldrich as the Senior Editor. The following individual involved in review of your submission has agreed to reveal their identity: Tibor Rohacs (Reviewer #2).

The reviewers have discussed the reviews with one another and the Reviewing Editor has drafted this decision to help you prepare a revised submission.

Summary:

The work by Harraz et al. investigates regulation of TRPV4 channel activity in brain capillary endothelial cells. The authors show that dialysis of the cells with hydrolyzable (Mg salt), but not with non-hydrolyzable (Na salt, ATP-γ-S) forms of ATP tonically suppresses TRPV4 currents. Using specific inhibitors of various protein and lipid kinases they demonstrate that the inhibition requires the activity of phosphatidyl inositol kinases, suggesting that it is mediated by PIP_2_. Indeed, cell dialysis with PIP_2_ causes TRPV4 inhibition even in the absence of Mg-ATP, and dialysis with the PIP_2_-scavenger polylysine reduces tonic inhibition by endogenous PIP_2_ even in the presence of ATP. Finally, they show that extracellular application of the G_q_PCR agonists PGE_2_ or carbachol potentiates TRPV4 activity, and this effect can be specifically abolished by pharmacological agents that inhibit individual components of the G_q_PCR-PLC pathway. Given the contradictory earlier reports of the effects of PIP_2_ on TRPV4 (activating vs. inhibiting), the use of complementary approaches is important. Another strength of the manuscript is that they study exclusively native cells, and use both TRPV4 inhibitors and a TRPV4 -/- mouse to demonstrate that the currents they measure are indeed mediated by TRPV4.

The data are of high quality, and support the authors' conclusions. Together with previous work by the authors (Harraz et al., 2018) the present work suggests that PIP_2_ depletion reciprocally regulates the hyperpolarizing Kir2.1 and the depolarizing TRPV4 current in brain capillary endothelial cells, identifying these as molecular players involved in the initiation and/or propagation of the vasodilation observed during functional hyperemia. All in all, this is a beautiful study which promotes mechanistic understanding of the process of neurovascular coupling in the brain, and is highly suitable for publication in *eLife*.

Essential revisions:

1) Experimental:

What is the natural activator of TRPV4 in brain cECs? Even upon PIP_2_ depletion whole-cell NP_o_ is only ~0.2 (amounting to a single-channel P_o_ of ~0.001), which is negligible channel activity. Thus, there must be a natural activator to induce TRPV4 currents in vivo. The authors use exclusively GSK101 as an agonist. In an earlier paper Garcia-Elias et al. (2013) showed that TRPV4 currents induced by heat and osmotic swelling (but not by 4aPDD) were inhibited by selective depletion of PIP_2_ using a chemically inducible lipid phosphatase. Given these contradictory results on positive vs. negative effects of PIP_2_ on TRPV4 channels it would be important to test whether the differential effects of PIP_2_ are due to different modes of activation, or to different cellular environments. e.g., are native TRPV4 channels activated by heat (Garcia-Elias used 38^o^C as stimulus), swelling, or epoxyeicosatrienoic acids (EETs; see Discussion, fifth paragraph) also suppressed by PIP_2_? Specifically, we would like to see experiments similar to those in Figure 5, but in the absence of GSK101, using a more "natural" way to activate TRPV4: e.g., body temperature+PGE_2_, or body temperature+EET – preferably both with and without diC8-PIP_2_ in the pipette.

2) Discussion:

In a previous study, the group demonstrated that extracellular K^+^ increases activate cEC Kir channels resulting in an upstream hyperpolarizing current which mediated capillary-to-arteriole dilation (Nat. Neurosci, 2017). In a follow up study, the authors showed that PIP_2_ is a molecular regulator of Kir2.1 channels and that G_q_PCR activation impairs capillary-to-arteriole electrical signaling by depleting PIP_2_ (PNAS, 2018). In the current study the idea that G_q_PCR signaling can regulate (via PIP_2_) the activation of opposing signaling pathways suggests that activation of these receptors stops neuronal activity-evoked changes in blood flow. However, this is contradictory to earlier evidence by Lacroix et al. (2015), demonstrating an important role for COX-2 derived PGE_2_ in the NVC response. The authors cite this work, but do not explain how their proposed mechanism relates to the published findings on PGE2 and/or other G_q_PCR agonists. Would all G_q_PCR impair or retard the upstream hyperpolarization initiated by K^+^? Would this pathway be more relevant during pronounced ischemia? Or is it relevant under physiologic conditions? A thorough investigation of this question is beyond the scope of the present work, but a more detailed discussion of unresolved issues would be appreciated.

---

## [Author Response]

Essential revisions:1) Experimental:What is the natural activator of TRPV4 in brain cECs? Even upon PIP_2_ depletion whole-cell NP_o_ is only ~0.2 (amounting to a single-channel P_o_ of ~0.001), which is negligible channel activity. Thus, there must be a natural activator to induce TRPV4 currents in vivo. The authors use exclusively GSK101 as an agonist. In an earlier paper Garcia-Elias et al. (2013) showed that TRPV4 currents induced by heat and osmotic swelling (but not by 4aPDD) were inhibited by selective depletion of PIP_2_ using a chemically inducible lipid phosphatase. Given these contradictory results on positive vs. negative effects of PIP_2_ on TRPV4 channels it would be important to test whether the differential effects of PIP_2_ are due to different modes of activation, or to different cellular environments. e.g., are native TRPV4 channels activated by heat (Garcia-Elias used 38C as stimulus), swelling, or epoxyeicosatrienoic acids (EETs; see Discussion, fifth paragraph) also suppressed by PIP_2_? Specifically, we would like to see experiments similar to those in Figure 5, but in the absence of GSK101, using a more "natural" way to activate TRPV4: e.g., body temperature+PGE_2_, or body temperature+EET – preferably both with and without diC8-PIP_2_ in the pipette.

We thank the reviewers for affording us the opportunity to address this question in some detail. The question itself requires a bit of unpacking:

1) “What is the natural activator of TRPV4 in brain cECs?”

Given that G_q_PCR signaling is the main mechanism for decreasing membrane PIP_2_ levels and our evidence that PIP_2_ depletion increases TRPV4 channel activity (~6-fold) in the absence of a TRPV4 agonist, we would argue that G_q_PCR agonists are the major physiological activators of TRPV4 in brain cECs. This aligns well with our ongoing studies using transgenic mice expressing a genetically encoded Ca^2+^ indicator, which show that G_q_PCR-stimulated TRPV4 channel activity is responsible for ~50% of the large, transient cEC Ca^2+^ signals induced by neural activity in vivo (Longden et al., World Congress of Microcirculation 2018). These data support the concept that low levels of TRPV4 channel activity induced by G_q_PCR agonists in the brain microenvironment can cause significant increases in intracellular Ca^2+^. Thus, although there may be endogenous direct activators of TRPV4 channels in the brain in addition to G_q_PCR agonists, they are not needed to support the physiological function of these channels.

2) “Even upon PIP_2_ depletion whole-cell NP_o_ is only ~0.2 (amounting to a single-channel P_o_ of ~0.001), which is negligible channel activity.”

The idea that a natural activator of TRPV4 currents in vivo is needed because NP_o_ is so low – even with PIP_2_ depletion – misses the very important point that even “negligible” TRPV4 activity is sufficient for this channel to exert its physiological functions. In fact, in a very real sense, mechanisms that *suppress* TRPV4 activity may be more important to the role of this channel in endothelial cells than those that activate it. Our previous studies highlight this point, demonstrating that exceedingly low levels of endothelial TRPV4 channel activity – equivalent to that produced by activation of 1–2 channels per cell – are capable of causing maximal dilation of systemic arteries (Sonkusare et al., 2012; 2014). Moreover, our computational modeling efforts show that the increase in TRPV4 current induced by G_q_PCR stimulation—from ~80 fA in the PIP_2_-suppressed state (at -40 mV) to ~400–500 fA with G_q_PCR stimulation – would effectively short circuit K^+^-induced hyperpolarization, reinforcing the importance of constraints on TRPV4 activation for the operation of this key neurovascular coupling (NVC) mechanism. Importantly, our results described in response to comment 1.1, above, also confirm that PIP_2_ depletion alone is sufficient to produce physiologically meaningful TRPV4 channel activation. We have clarified these quantitative aspects in the revised manuscript.

3) “Is PIP_2_ required for some modes of channel activation?”

As the reviewers indicate, one group (Suetsugu and colleagues, Nat Commun) reported that PIP_2_ is inhibitory towards all modes of activation, whereas another group (Valverde and colleagues, PNAS) reported that PIP_2_ is required for heat-, hypotonicity- and epoxyeicosatrienoic (EETs)-induced activation, but not for 4-αPDD–induced activation. Our results indicate that, in the absence of any agonist, PIP_2_ depletion increases TRPV4 channel open probability about 6-fold. In the presence of either of two distinct synthetic agonists, GSK101 or 4-αPDD (new supplementary figure), PIP_2_ depletion also increases TRPV4 channel activity about 6-fold. In response to the reviewers’ comment, we tested the purported physiological activator of TRPV4 channels, 11,12-EET, and found that it activated TRPV4 currents even in the absence of any dialyzed PIP_2_ (Figure 3—figure supplement 2 in the revised manuscript), a result that contrasts with that of Garcia-Elias et al. (2013). We chose 11,12-EET because it has been implicated in NVC and is thought to transduce the effects of osmolarity (Garcia-Elias et al., 2013). (We did not test heat in excised patches because giga-ohm seals on cECs are unable to withstand temperature elevation.) Collectively, our results support the concept that PIP_2_ suppresses TRPV4 channel activity independent of the mode of activation, and are congruent with the results of Takahashi et al., (2014).

2) Discussion:In a previous study, the group demonstrated that extracellular K^+^ increases activate cEC Kir channels resulting in an upstream hyperpolarizing current which mediated capillary-to-arteriole dilation (Nat. Neurosci, 2017). In a follow up study, the authors showed that PIP_2_ is a molecular regulator of Kir2.1 channels and that G_q_PCR activation impairs capillary-to-arteriole electrical signaling by depleting PIP_2_ (PNAS, 2018). In the current study the idea that G_q_PCR signaling can regulate (via PIP_2_) the activation of opposing signaling pathways suggests that activation of these receptors stops neuronal activity-evoked changes in blood flow. However, this is contradictory to earlier evidence by Lacroix et al. (2015), demonstrating an important role for COX-2 derived PGE_2_ in the NVC response. The authors cite this work, but do not explain how their proposed mechanism relates to the published findings on PGE2 and/or other G_q_PCR agonists. Would all G_q_PCR impair or retard the upstream hyperpolarization initiated by K^+^? Would this pathway be more relevant during pronounced ischemia? Or is it relevant under physiologic conditions? A thorough investigation of this question is beyond the scope of the present work, but a more detailed discussion of unresolved issues would be appreciated.

The question of whether G_q_PCR activation, particularly by PGE_2_, mediates NVC or serves to interrupt neuronal activity-evoked changes in blood flow, as speculated here, reveals limitations in our current understanding of how various features of neuronal signaling to the vasculature interact to create the meta phenomenon of activity-dependent control of blood flow distribution, as reflected in BOLD-fMRI measurements. On theoretical grounds, our current and previous (Harraz et al., 2018) findings support the idea that G_q_PCR signaling in capillary endothelial cells impedes electrical signaling through PIP_2_-depletion–mediated deactivation of Kir2.1 channels and enhancement of depolarizing currents through TRPV4 channels. In this sense, local G_q_PCR signals would act as “stop signs” in the pathway of electrical signaling, and could thereby conceivably sculpt the extent and directionality of signaling. However, we also have evidence that direct application of PGE_2_ to capillary extremities in an ex vivo capillary-parenchymal arteriole preparation acts via an as-yet unidentified mechanism to cause dilation of the upstream arteriolar segment, consistent with a contribution of PGE_2_ to NVC at the capillary level (Dabertrand, Brayden & Nelson, 2015 - Society of General Physiologists Symposium, abstract published in J Gen Physiol 146(3): 264). (As we reported previously (Dabertrand et al. J Cereb Blood Flow Metab. 2013; 33(4):479-82), direct application of PGE_2_ to arterioles causes arteriolar constriction.) Thus, we would suggest that these two functions – NVC mediator and “stop sign” - are not necessarily mutually exclusive. We would also note that G_q_PCR activation would lead to an elevation of capillary endothelial cell [Ca^2+^]_i_, which should activate nitric oxide (NO) synthase to cause the production of NO, providing an additional mechanism by which PGE_2_ signaling might relax adjacent smooth muscle or contractile pericytes and contribute to NVC.

As indicated in Figure 5—figure supplement 2 in the original manuscript, the G_q_PCR agonist carbachol exerts an effect comparable to that of PGE_2_, whereas the PAR-2 ligand SLIGRL and the purinergic receptor agonist ATP had no effect. The absence of an effect of SLIGRL may reflect the absence of PAR-2 in brain capillary endothelial cells, as suggested by a previous transcriptomic study (Vanlandewijck et al., 2018). The inability of ATP to enhance TRPV4 channel activity is more difficult to explain, but is apparently not attributable to the absence of an appropriate P2Y receptor in capillary endothelial cells since ATP is capable of inhibiting Kir currents in these cells (Harraz et al., 2018). Thus, although there is some degree of generalizability, there may be more to the story than the simple ability to signal through a G_q_PCR.

In terms of the physiological context in which this pathway operates, our working assumption is that this stop sign-like function acts to shape Kir-mediated electrical signaling under physiological conditions. We would further predict that ischemia acts through a reduction in intracellular ATP to favor TRPV4 signaling over Kir2.1 signaling.